

# A Synthesis of Space-time Variability in Multi-Component Flood Response

Yiwen Mei, Xinyi Shen and Emmanouil N. Anagnostou

Civil and Environmental Engineering, University of Connecticut, Storrs, CT, USA

*Correspondence to:* Emmanouil N. Anagnostou (manos@engr.uconn.edu)

**Abstract.** Catchment flood response consists of multiple components of flow generated by the heterogeneity of catchment surface. This study proposes an analytical framework built upon the Viglione et al. (2010a) to assess the dependence of catchment flood response on different flow components. The analytical framework is compared to simulations from a distributed hydrologic model. A large number of rainfall-runoff events from three catchments of Tar River basin in North Carolina are used to illustrate the analytical framework. Specifically, the framework is used to estimate three flood events characteristics (cumulative runoff volume, centroid and spreadness of hydrograph) through three corresponding framework parameters: the rainfall excess and the mean and variance of catchment response time. Results show that under the smooth topographic setups of the study area, the spatial and/or temporal correlation between rainfall and runoff generation are insignificant to flood response; delay in flood response due to runoff generation and routing are of equal importance; the shape of flood is mainly controlled by the variability in runoff generation stage but with non-negligible contribution from the runoff routing stage. Sensitivity tests show that the framework's main error source is the systematic underestimation of flood event's centroid and spreadness, while the random error is relatively low.

## 1. Introduction

Catchment flood response or, in a more general sense, the water balance at basin scale, is controlled by a range of hydrological processes with each of them contributing a different level of spatiotemporal variability (e.g. precipitation, surface runoff, infiltration, routing, etc.) (Skøien, et al., 2003; Skøien & Blöschl, 2006; Merz & Blöschl, 2009; Rodríguez-Blanco, et al., 2012; Palleiro, et al., 2014; Zoccatelli, et al., 2015). Many of these studies have investigated how these processes are linked with the catchment flood response and what the relative importance of these processes are in controlling the properties of flood being generated. For example, it has been argued that only a portion of space-time characteristics of the flood response process will emerge to control the dynamics of a flood hydrograph due to the catchment dampening effect (Skøien, et al., 2003; Smith, et al., 2004; Skøien & Blöschl, 2006), and this dampening effect varies dynamically according to the hydrogeological properties of the catchment and features of the triggering storm, implying a shift of relative importance of processes in catchment flood response under different flood regimes (Sivapalan, et al., 2004; Smith, et al., 2002; 2005; Sangati, et al., 2009; Mejía & Moglen, 2010; Volpi, et al., 2012; Mei, et al., 2014). The answers to these questions are intimately related to the development of a comprehensive framework that can generalize the estimation of streamflow spatiotemporal variability by a synthesis of various catchment processes under different hydro-meteorological and geomorphological controls (Blöschl, 2006).

Describing catchment flood response based on a set of spatiotemporal variables in storm response (i.e. rainfall, runoff generation, and routing) has been established and utilized since the late 90s (Woods & Sivapalan, 1999; Smith, et al., 2005; Viglione, et al., 2010a; Mejía & Moglen, 2010; Mei, et al., 2014; Zoccatelli, et al., 2011; 2015). The essence of such an analytical framework is to diagnose the relative importance of rainfall space-time processes





in influencing the runoff generation (i.e. cumulative flow volume, hydrograph timing and shape). The first work that synthesized the space-time variables into a holistic analytical framework is that of Woods & Sivapalan (1999). That framework used the "stationary rainfall" assumption, which can be interpreted as no movement of rainfall over the catchment. This assumption is strong, but it is considered reasonable only for short-duration or orographic-enhanced storms, which have relatively fixed spatial patterns over time. This framework assumption was applied in subsequent studies by Mejía & Moglen (2010) and Zoccatelli et al. (2010). Specifically, Zoccatelli et al. (2010) investigated the influence introduced by neglecting the spatial information of rainfall distribution in flow generation. Their study showed a larger delay in the arrival of a hydrograph mass center as rainfall mass center tends to be located closer to the headwater of the basin. Mejía & Moglen (2010) investigated the flood response to the distribution of impervious surface by partitioning rainfall excess generation to pervious and impervious areas of a catchment. The study concluded that the impervousness pattern is important when it is collocated with the mass center of rainfall.

Viglione et al. (2010a) generalized the Woods & Sivapalan (1999) framework by relaxing the "stationary rainfall" assumption. Their framework has terms to describe the relative movement of rainfall to the other variables. Viglione et al. (2010b) utilized their generalized framework to study the relative importance among rainfall space-time processes in controlling runoff generation for different types of flood. The study points out that the space and time covariance are important in runoff generation for short-duration rainfall events due to their highly localized feature; the spatial covariance is irrelevant for long-duration rainfall events since the rainfall field tends to be uniformly distributed over the catchment. Zoccatelli et al. (2011) derives the spatial moment of catchment rainfall and catchment scale storm velocity under the constant runoff coefficient assumption. The results indicate that the closer the rainfall mass center is to the catchment outlet the earlier the arrival of the hydrograph mass center is. This aspect was also revealed in Mei et al. (2014), which examined 164 (mostly moderate) flood events. The study further concludes that the shape of rainfall and its movement are relatively insensitive in shaping the event hydrograph mainly because of the unsaturated rainfall excess. Nikolopoulos et al. (2013) paid particular attention to the catchment scale storm velocity and were able to demonstrate the scale dependency and rainfall intensity dependency to storm magnitude.

The Viglione et al. (2010a) analytical framework (hereafter referred to as V2010) is relevant to only one rainfall excess (event flow) component. From this sense, the different flood responses due to the highly heterogeneous land surface properties within a catchment are lumped together (Woods & Sivapalan, 1999; Viglione, et al., 2010b). Numerous experimental studies have demonstrated that catchment flood response can be identified as multiple components (Weiler, et al., 2003; Liu, et al., 2004; Gonzales, et al., 2009). This is also described from the hydrologic modeling perspective; rainfall excess is often partitioned into different linear reservoirs representing different routing mechanisms (Koren, et al., 2004; Blöschl, et al., 2008; Wang, et al., 2011). Thus, we see the necessity of further generalizing V2010 to represent multi-component flood responses. The analytical framework presented in this paper is visualized in Figure 1. Catchment rainfall forcing is converted to more than one rainfall excess components by the various land cover types. These rainfall excess components are subjected to different routing schemes associated with the different land cover classes. The output hydrograph is a combination of hydrographs from all components. A point to note is that our discretization of streamflow is still within the context of event flow and is not extended to the very slow response (e.g. baseflow). To sum up, our expanded framework introduces parallel channels to represent the different components of catchment flood response compared to V2010 and any previous studies regarding this topic. This can facilitate our understanding of which space-time process is the most dominated for a component of catchment response and how the contributions of different rainfall excess



components are changing across disparate hydrologic regimes (e.g. basin scales, rainfall duration, and space-time distribution, etc.).

We illustrate the multi-component flood response framework based on a two-component assumption consistent with a distributed hydrologic model structure. The illustration is built based on a relatively large number of rainfall-runoff events from three catchments in the Tar River basin in North Carolina. The paper is organized as follows. In Section 2, the study basin and data used in the study are described. Section 3 illustrates the experimental design with the hydrologic model. The analytical framework equations together with the demonstrations are presented in Section 4. Tests to understand the framework sensitivity to flood characteristics are provided in section 5. Conclusions (including limitation and future works) are discussed in section 6.

## 2. Study Area and Dataset

### 2.1. Tar River Basin and Hydrometeorology Data

We conducted our analysis over three nested catchments (namely Swift, Fishing and Tar) in the Tar River basin, a low elevation basin located in North Carolina (maximum elevation is 220 m above sea level). The study catchment areas are 426 km$^2$, 1374 km$^2$, and 2406 km$^2$, characterized by mild-slopes (mean slope at 0.90%, 0.81%, and 0.83%, respectively). Prevailing climate of the area is humid subtropical causing annual precipitation and runoff around 1100 mm and 250 mm, respectively. The readers are encouraged to read Mei & Anagnostou (2015) for details on the hydrology of the study area.

The Stage IV radar-based multi-sensor precipitation estimates from the National Center for Environmental Prediction is used as our reference rainfall (STIV hereafter). The product is mosaicked from the Regional Multi-sensor Precipitation Analysis (RMPA) produced by the National Weather Service River Forecast Centers and benefits from some manual quality control process (Lin & Mitchell, 2005). The RMPA includes rain rates from merged operational radar estimates (150 Doppler Next Generation Weather Radar) and 5500 hourly rain gauge measurements. The STIV data is hourly and available at approximately 4 km spatial resolution. The data used in this study has been spatially interpolated to 1 km by the bilinear method. Another meteorology data is the potential evapotranspiration (PET) data available from the North American Regional Reanalysis (NARR) at 3-hourly and 32 km resolution (Mesinger, et al., 2005). The NARR PET product accounts for evaporation from the soil, transpiration from the vegetation canopy, evaporation of dew and frost or canopy-intercepted precipitation, and snow sublimation. The hourly flow rate is aggregated from the 15-min flow rate record available from Unite States Geology Survey (USGS) for the three study catchments.

### 2.2. Rainfall-runoff Event Selection

The study rainfall-runoff events are extracted from the observation datasets using the Characteristic Point Method (CPM) introduced in Mei & Anagnostou (2015). The advantages of CPM are its parsimony data requirement (basin area and time series of rainfall and flow) and automatic extraction of events based on time series features. Event runoff and rainfall periods are identified from the long-term continuous time series of observed flow and rainfall records. Rainfall periods satisfying the following conditions are associated with each of the flood periods:
- rainfall period(s) occurring before the flood period but within the time of concentration of the basin;
- rainfall period(s) located on the rising lib of the flood period;



- rainfall period(s) occurring prior to the end of the flood period by a time length equal to the time of concentration.

All of the rainfall periods associated with the same flood period are integrated as one rainfall event and are considered as the inducing rainfall of the flood. Each of the rainfall and flood pairs forms a rainfall-runoff event. The CPM is applied on the USGS observed flow rate and catchment-average STIV rainfall data of the three study catchments. The method resulted in nearly 300 events from the study years and these events were further filtered according to the hydrologic model performance which is described in section 3.2.

## 3. Hydrologic Model and Experiment

### 3.1. Distributed Hydrologic Model

The Coupled Routing and Excess STorage (CREST) model version 2.1 is used for the hydrologic simulations in our study. CREST is a fully distributed rainfall-runoff model designed to simulate flow discharges over watersheds at global scale. CREST integrates a water balance model for the vertical fluxes with a horizontal routing model for the surface and subsurface runoff (Wang, et al., 2011; Shen, et al., 2015). The water balance model considers four processes—canopy interception, infiltration, evapotranspiration (ET), and runoff generation. The infiltration rate is calculated based on the variable infiltration curve developed in the Xinanjiang Model (Zhao, 1992). For each grid-cell, the actual ET (AET) is determined in terms of water and energy budget using precipitation, soil water availability and potential ET. In the runoff generation process, CREST separates rainfall excess into two components, the surface and subsurface runoff modeled by two linear reservoirs—the overland and interflow reservoirs. In the routing process, the sub-grid routing inhomogeneity is accounted for by employing these two runoff components. The model version we are using implements the fully distributed linear reservoir routing method (Shen, et al., 2015) that overcomes the severe underestimation of flow in previous versions. The parameter-optimization algorithm adapted in CREST is the shuffled complex evolution (SCE-UA) developed by Duan et al. (1992).

### 3.2. Experimental Design

As a first step, the model is set up over the three study catchments (Shen & Hong, 2015). The geomorphologic and hydrologic variables (i.e. flow direction, flow accumulation, slope, and stream channel) of the catchment areas are generated from the Digital Elevation Model (DEM) data; the STIV precipitation and NARR PET product force the model to compute the through precipitation, actual evapotranspiration, infiltration capacity, soil water content, and rainfall excess. We keep the model setting relatively simple by "turning off" the canopy interception, meaning that the process is conceptualized by a multiplier of the precipitation data, which is optimized by a calibration process. The model was calibrated in the three catchments with respect to the observed hourly flow rate from 2004 to 2006 (year 2002 to 2003 is used as the spinning period). The Nash-Sutcliff coefficient (*NSCE*) of the flow simulations in the Swift, Fishing, and Tar catchments determined at hourly scale are 0.69, 0.62, and 0.66, respectively, indicating reasonable performance of the model over the study catchments (Moriasi, et al., 2007).

Rainfall-runoff events from the 2003 to 2012 period with duration shorter than 500 hours are identified from the continuous flow simulations during the time periods provided by the CPM. The mean error (*ME*), correlation coefficient (*CC*), and *NSCE* are calculated with respect to the observed flow rate for each event and these error metrics are ranked in ascending order. Flood events of *ME* higher than the 95[th] percentile and *CC* and *NSI* lower





than their respective 5th percentile were discarded from the analysis to keep our results representative in the context of hydrologic simulation. These selection criteria resulted in 180 events (62, 57, and 61 events, respectively, for the smallest to largest catchments) with overall relative centered root mean square error equal to 42.0%, 43.0% and 34.4%, respectively. Two pilot events used in the framework demonstrations are exhibited in Figure 2. The first is an intermittent event lasting for 93 hours from June 2006; the second one is a 66-hour long event from November 2009. It can be clearly seen that the concentration time is generally longer for basin of a larger drainage area. Rainfall mass triggering the June 2006 event is distributed around the outlet while the November 2009 one is spatially bimodal.

In the last step, we introduce a method to remove the influences of non-zero initial condition of each event in the continuous simulations. For each event period, we run CREST by setting both rainfall and PET data to zero so as to output the "baseflow hydrograph". This "baseflow hydrograph" gives or mimics the recession of flow with the initial condition over the event time period. The event flow hydrographs are subtracted by the baseflow hydrographs and the new event flow hydrographs are obtained for the subsequent analysis.

## 4. Analytical Framework of Catchment Response

The V2010 analytical framework quantifies the effects of spatiotemporal variability of rainfall, direct runoff, and flow on flood response. The follow-up application by Viglione et al. (2010b) isolated the quick flow component from total flow simulated using the Kamp model and demonstrated the magnitudes of terms representing the different catchments of space-time process of the V2010. In our study, we extend the V2010 framework with the consideration that event flow consists of multiple components as shown in Figure 1. In this paper we illustrate the new framework setting the "multiple components" to two—surface and subsurface processes. This is consistent with the modeling structure of the hydrologic model (CREST). Similar to V2010, our analytical framework estimates catchment response by three quantities: a) the amount of rainfall excess, b) the mean catchment response time, and c) the variance of catchment response time. These three quantities are estimators of the corresponding flood characteristics, namely, a) the cumulative event flow volume, b) the hydrograph centroid, and c) the spreadness of hydrograph. A two-stage framework structure that decomposes the catchment response to rainfall excess generation and runoff routing is adopted (Zoccatelli, et al., 2011; Mei, et al., 2014). In this section, we focus on deriving the analytical framework equations under the multi-component scenario (sections 4.2 and 4.3). The variables used in the framework (rainfall, runoff coefficient, and runoff routing time) are attained from the CREST model parameters and are described in section 4.1.

### 4.1. Analytical Framework Variables

The analytical framework has three input variables—the rainfall, the runoff coefficient, and the runoff routing time. The rainfall variable for the framework refers to the net amount of through-rainfall and its partition in surface and subsurface runoff as defined in CREST. Thus, the effects of vegetation interception and AET are not considered in this study. Based on this definition, rainfall variable, $P(a,t)$, for the analytical framework is defined as:

$$P(a, t) = C_I P'(a, t) - E_a(a, t) \tag{1}$$

where $P'(a,t)$ and $E_a(a,t)$ are the input rainfall variable (i.e. the STIV data) and the actual evapotranspiration rate (provided by CREST in this study). Indexes $a$ and $t$ stand for the location and time dimensions. $C_I$ is the multiplier that conceptualizes the canopy.





Most distributed hydrologic models separate the rainfall excess into two components—the surface and the subsurface rainfall excess—and route them by two parallel flow paths with different speeds and outflow rates. Namely, the flood response for surface and subsurface components are associated with different generations and routing mechanisms characterized by different runoff coefficients and runoff routing time. The surface process is intimately related to the percentage of impervious surface over the basin where the through-rainfall is not infiltrated. As the percentage of impervious surface is modeled by the impervious ratio in CREST, the surface runoff coefficient, $W_2(a,t)$, is modeled in the proposed framework by the imperviousness, $I_M$, which is considered constant for a given basin. Values of $I_M$ for the three catchments (from small to large) are 13.1%, 10.9%, and 11.3%. On the other hand, the flow response through the subsurface process is correlated to the soil water capacity. Thus, the subsurface runoff coefficient, $W_1(a,t)$, is estimated as:

$$W_1(a,t) = \frac{SM(a,t)}{W_M} \tag{2}$$

where $SM(a,t)$ is the volumetric soil moisture (one of the model outputs), and $W_M$ is the mean water storage calibrated from the model.

For runoff routing, CREST modeled the basin surface as hillslope and channel grids with different concentration times. The concept of concentration time is a measurement of the required time for the rainfall excess to drain from its originating grid to the next downstream grid. Travelling time of water from a given grid-cell is calculated as the summation of all the concentration time along its flow path to the basin outlet. Therefore, the runoff routing time is written as:

$$\Theta(a) = \frac{\alpha}{K} \sum_{L(a)} \frac{l(a)}{s(a)^\beta} \tag{3}$$

where $l(a)$ and $s(a)$ are the length of flow path from a grid to its adjacent downstream grid and the slope at that grid, respectively; $L(a)$ represents the flow path from a grid-cell to the catchment outlet; $K$ is the overland runoff velocity coefficient varies between hillslope and channel grids and $\beta$ is the overland flow speed exponent. The term $\alpha$ is a velocity coefficient used to distinguish the surface and subsurface routing. In our study, $\alpha$ is unity in Eq.(3) for the subsurface runoff routing time, $\Theta_1(a)$; it takes values smaller than one to represent $\Theta_2(a)$, the surface runoff routing time since surface routing should be faster than subsurface routing. Values of $\alpha$ are 0.31, 0.63 and 0.67 for the study catchments. Therefore, routing times for the surface and subsurface processes are proportional to each other:

$$\frac{\Theta_2(a)}{\Theta_1(a)} = \alpha \tag{4}$$

Magnitudes of the spatially variable runoff routing time for the surface and subsurface processes are illustrated in Figure 3. The figure shows that runoff routing time increases going upstream. The Tar catchment (the largest one) is characterized by the widest value range compared to the other two sub-basins. The Swift catchment shows distinctively lower overall values for the surface runoff routing time ($\Theta_2$) due to its low $\alpha$ value (around 0.3 compared to 0.65 for the other two). This is expected given that model parameters of the three catchments are independently calibrated.

## 4.2. Generation of Rainfall Excess

The generation of rainfall excess is calculated using the parsimonious rainfall-runoff equation following V2010:

$$R_i = PW_i \tag{5}$$





where $R_i$, $P$ and $W_i$ are the space-time variable rainfall excess, precipitation, and runoff coefficient field. Index $i$ indicates different rainfall excess components. In this study $i$ is 1 or 2 to denote the subsurface and surface rainfall excess, respectively. Following mass balance, we write the total rainfall excess under our multiple flow component assumption as:

$$R = P \sum_{i=1}^{N} W_i \tag{6}$$

Note that the sum of all $W_i$ is the total runoff coefficient $W$. To calculate the rainfall excess generated during a storm event over a catchment, we integrate Eq.(6) over the catchment area or storm period:

$$[R]_a = [P]_a \sum_{i=1}^{N} [W_i]_a + \left\{ P, \sum_{i=1}^{N} W_i \right\}_a \tag{7}$$

$$[R]_t = [P]_t \sum_{i=1}^{N} [W_i]_t + \left\{ P, \sum_{i=1}^{N} W_i \right\}_t \tag{8}$$

where $[*]$ and $\{*\}$ with subscript $a$ or $t$ stand for the expectation and covariance (variance if the variables are the same) operator applied over catchment area or storm period, respectively. The first term in Eq.(7)/(8) is the product between spatial or temporal average rainfall and runoff coefficient, while the second term quantifies the spatial or temporal variability between rainfall and runoff coefficient at every time step/catchment grid.

Figure 4 and Figure 5 show the magnitudes of the different terms of Eq. (7) and (8), respectively. Note that since the surface runoff coefficient $W_2$ is estimated as a space-time constant, the space and time covariance term between $W_2$ and $P$ (the 2nd term in Eq.(7) and (8)) are 0 and are not shown. It is observed from Figure 4 that the $[R]_a$ time series preserves the temporal variability of catchment-average rainfall (shown in Figure 2); the spatial covariance term between $P$ and $W_1$ is irrelevant to $[R]_a$ compared to the product term ($[P]_a[W_1]_a$). This indicates that rainfall and runoff coefficients are not collocated in space for the two pilot events. Meanwhile, the relative importance between $[P]_a[W_1]_a$ and $[P]_a[W_2]_a$ changes dynamically through the event where $[P]_a[W_1]_a$ and $[P]_a[W_2]_a$ are comparable during the early phase but $[P]_a[W_1]_a$ overwhelms the other in the mature and decade phase of the event. To further investigate this aspect, we plot the differences between $[W_1]_a$ and $[W_2]_a$ of the three catchments separately in Figure 6's top two panels. As expected, the differences start at negative and then change to positive during the evolution of the event, reflecting larger $[P]_a[W_2]_a$ and later $[P]_a[W_1]_a$. This dynamic change in $[W_1]_a$ and $[W_2]_a$ also demonstrates why the surface rainfall excess component is the quick response from the model. In addition, the differences between $[W_1]_a$ and $[W_2]_a$ of the Swift catchment are noticeably larger than the other two catchments in Figure 6; this is attributed to the large gap between $[P]_a[W_1]_a$ and $[P]_a[W_2]_a$ of the Swift catchment shown in Figure 4.

Figure 5 illustrates the temporal aggregated maps for terms in Eq.(8). The product terms between temporal average rainfall and runoff coefficient ($[P]_t[W_1]_t$ & $[P]_t[W_2]_t$) account for the major contribution of $[R]_t$. $[P]_t[W_1]_t$ is generally larger than $[P]_t[W_2]_t$ because of $[W_1]_t$ which is of larger values than $[W_2]_t$ as shown in the bottom two panels of Figure 6. This is particularly seen for the smallest Swift catchment cases. For the subsurface rainfall excess, its temporal covariance term ($\{P,W_1\}_t$) is irrelevant except for the June 2006 event in the Swift catchment. This implies that there is little correlation between rainfall dynamics and the subsurface runoff coefficient during the event period over most catchment locations.

The temporal or spatial integration of Eq.(7)/(8) yields the catchment-average storm rainfall excess, $[R]_{at}$:





$$[R]_{at} = \underbrace{[P]_{at}\sum_{i=1}^{N}[W_i]_{at}}_{R1} + \underbrace{\left\{[P]_a, \sum_{i=1}^{N}[W_i]_a\right\}_t}_{R2} + \underbrace{\left\{[P]_t, \sum_{i=1}^{N}[W_i]_t\right\}_a}_{R3}$$
$$+ \underbrace{\left[\left\{(P-[P]_t), \sum_{i=1}^{N}(W_i-[W_i]_t)\right\}_a\right]_t}_{R4} \tag{9}$$

where $[*]_{at}$ is the space-time aggregation on the catchment area and event period. This equation indicates that the amount of total catchment-average storm rainfall excess is the sum of catchment-average storm rainfall excess from all components. Eq.(9) corresponds also to Eq.(9) in V2010, but it is a more general form since more than one flow components are considered in the equation. Meanings of terms in Eq.(9) are consistent with those of Eq.(9) in V2010 except that the current Eq.(9) is compatible for more than one flow components. Moreover, V2010 has shown that the effect of storm movement can be isolated as *R4-R2·R3/R1*. This movement effect is also studied later.

The magnitudes of terms in Eq.(9) along with the movement effect, *MV*, for the study events are illustrated in Figure 7 with statistics summarized in Table 1 (the two sample events are highlighted in the figure). Note that the *R2*, *R3*, *R4*, and *MV* for the surface component are zero due to the space-time constant surface runoff coefficient and thus are not shown. A term-wised comparison shows clearly that *R1* is the most dominant contributor to $[R]_{at}$. The figure and table reveal that the spatial and temporal correlation between rainfall and runoff coefficients is almost negligible. This is consistent with the previous studies which show generally low magnitudes of the *R2*, *R3*, and *R4* but high $R_1$ (Viglione, et al., 2010b; Mejía & Moglen, 2010). The relatively low magnitudes of term $\{P,W_1\}_a$ and $\{P,W_1\}_t$ in Figure 4 and Figure 5 also agree with this observation. The fairly low magnitudes of space and time covariance lead to insignificant movement effect (mean at $10^{-3}$ mm/h from Table 1). Investigation on *R1* (the most significant term) shows a decrease in magnitude with basin scale. This dampening effect has different reasons for the two rainfall excess components. For the surface one, diminishing in magnitude with increase in scale is a result of the decrease in catchment-average rainfall given that $W_2$ are constant among catchments. For the subsurface process, this is due to both the decrease in runoff coefficient and catchment-average rainfall. Moreover, Table 1 reveals that the subsurface component generally outperforms surface one in contribution to *R1*. Yet this magnitude differences are diminishing from the smallest to the largest catchment since the gap between $W_1$ and $W_2$ is narrowing.

### 4.3. Catchment Response Time

The catchment response is conceptualized by two stages—rainfall on the catchment and a portion of it turning into rainfall excess; then the rainfall excess is routed to the catchment outlet (Zoccatelli, et al., 2011; Mei, et al., 2014). These two stages are associated with their own "holding times" which are treated as random variables (Rodríguez-Iturbe & Valdés, 1979). The catchment response time measures the time needed from the beginning of a storm to a drop of rainwater exiting the catchment outlet, whose probability distribution function (PDF) for the *i*-th rainfall excess component, $f_{Ri}$, is

$$f_{R_i} = \frac{R_i}{[R_i]_{at}} \tag{10}$$

Note that $f_{Ri}$ is a space-time variable. Thus, the PDF for total rainfall excess, $f_R$, can be written as:

$$f_R = \sum_{i}^{N} \psi_i f_{R_i} \tag{11}$$





where $\psi_i$ is the rainfall excess ratio defined as the ratio of catchment-average storm rainfall excess for a component to that for the total rainfall excess:

$$\psi_i = \frac{[R_i]_{at}}{[R]_{at}} \tag{12}$$

Sum of $\psi_i$ goes up to 1 by definition. Eq.(11) shows that the PDF of catchment response time is a convex combination for each distribution of the rainfall excess component.

### 4.3.1. Expectation of Catchment Response Time

For the two-stage analytical framework in this study, the expectation of catchment response time $E(\Phi)$ can be decomposed to the expectation of holding time of the two stages:

$$E(\Phi) = \underbrace{E(T)}_{Stage1} + \underbrace{E(\Theta)}_{Stage2} \tag{13}$$

where $T$ and $\Theta$ correspond to the rainfall excess generation time and runoff routing time. Following V2010, the expected rainfall excess generation time, $E_i(T)$, for any component is provided as:

$$E_i(T) = \frac{|T_P|}{2} + \frac{\{T, [R_i]_a\}_t}{[R_i]_{at}} \tag{14}$$

where $|T_P|$ is the duration of rainfall event. $E_i(T)$ is a measurement of the temporal mass center of rainfall excess. If the rainfall mass is symmetric with respect to its mid-point, the half-duration is sufficient to describe the expectation of rainfall excess generation. Following the distribution function of Eq.(11), we derived the expected rainfall generation time for total rainfall excess $E(T)$ as (see Appendix I for the derivation):

$$E(T) = \sum_i^N \psi_i E_i(T) \tag{15}$$

Eq.(15) indicates that the temporal mass center of total rainfall excess is a linear combination (or the expectation) of the mass centers of all the other rainfall excess components with respect to the rainfall excess ratio. The equation also implies that the larger the magnitude of a component, the greater impact it has on the timing of the total rainfall excess. Substituting Eq.(14) into Eq.(15), we have,

$$E(T) = \underbrace{\frac{|T_P|}{2}}_{E1} + \underbrace{\frac{\{T, \sum_i^N [R_i]_a\}_t}{[R]_{at}}}_{E2} \tag{16}$$

Eq.(16) is a more general form of Eq.(15) in V2010 with the consideration of more than one rainfall excess component.

The magnitudes of terms in Eq.(16) are illustrated in Figure 8 (left panel) and summarized in Table 2 (first three rows). At a first glance, the expectation of catchment response time is increasing with the basin area due to the increase in event duration. The magnitude of the half-duration is of more relevance to the expectation of catchment response time while the temporal covariance term can be an important contributor for a portion of events. This means that rainfall excess is not symmetric with respect to the event's mid-point. *E2* of the surface component is higher than the subsurface counterpart in magnitude. This is interpreted to mean that the surface rainfall excess preserves the temporal dynamics of catchment-average rainfall due to the constant runoff coefficient. On the other hand, for the subsurface component, the temporal characteristics of rainfall have been dampened through its interaction with runoff coefficients. This leads to a more symmetrically distributed time series based on the mid-



point. Besides, Table 2 implies that the temporal locations of rainfall excess mass center appear earlier than the event's mid-point by rendering negative mean *E2*. Lastly, we observe that the *E2* term of the June 2006 event is characterized by a larger value than the other events. This can be interpreted by its increasing trend in rain rate with time exhibited in the time series of Figure 2.

Holding time for the second stage is modeled by the spatial variable runoff routing time ($\Theta_i$) detailed in section 4.1. The expectation of the runoff routing time for the rainfall excess component is derived as:

$$E_i(\Theta_i) = [\Theta_i]_a + \frac{\{\Theta_i, [R_i]_t\}_a}{[R_i]_{at}} \tag{17}$$

This equation does not correspond to a single equation in V2010 since the hillslope and channel routing process are lumped as one runoff routing process in our assumption. The first term stands for the catchment-average runoff routing time and the second term quantifies the delay in response due to spatial covariance between runoff routing

time and storm-average rainfall excess. Analogously, we describe the relationship between $E_i(\Theta_i)$ and $E(\Theta)$. As a first step, an analytical relationship linking $\Theta_i$ and $\Theta$ together is required. Knowing that runoff routing time for the total rainfall excess should be between those for the slowest and fastest components, we assume $\Theta$ is a linear combination of all $\Theta_i$ with respect to the rainfall excess ratio $\psi_i$:

$$\Theta = \sum_i^N \psi_i \Theta_i \tag{18}$$

Under such an assumption, $\Theta$ neither goes beyond nor below the slowest and quickest responses. Also, we simply

assume that the ratio between each two $\Theta_i$ is a space-time constant:

$$\alpha_i = \frac{\Theta_i}{\Theta_1} \tag{19}$$

This is consistent with the CREST model scheme as shown in Eq.(4). Based on Eq.(19), $\alpha_1$ is always 1 and $\alpha_2$ is the $\alpha$ in Eq.(4). From Eqs.(18) & (19), we may further write:

$$\Theta = \xi_i \Theta_i \tag{20}$$

where

$$\xi_i = \frac{1}{\alpha_i} \sum_i^N \psi_i \alpha_i \tag{21}$$

$\xi_i$ is the ratio between the weighted average of $\alpha_i$ (with respect to $\psi_i$) and $\alpha_i$. It is a measure of disparity in routing

time from a rainfall excess component to the total one. It accounts for the hydrologic and geomorphologic effects as the inclusion of $\psi_i$ and $\alpha_i$. With Eq.(20), the expectation of $\Theta$ is derived as (see Appendix II):

$$E(\Theta) = \sum_i^N \psi_i E_i(\xi_i \Theta_i) \tag{22}$$

Mathematically, Eq.(22) indicates that $E(\Theta)$ is the expectation of $E_i(\xi_i\Theta_i)$, but not $E_i(\Theta_i)$, with respect to $\psi_i$. Eq.(22) implies that both the hydrologic and geomorphologic effects are accounted for by combining all the $E_i(\Theta_i)$ in $E(\Theta)$. Note that $\xi_i$ may be pulled out from the expectation since it is neither a space nor a time variable. Substituting

Eq.(17) into Eq.(22), $E(\Theta)$ may be written as:

$$E(\Theta) = \underbrace{\sum_i^N \psi_i \xi_i [\Theta_i]_a}_{E3} + \underbrace{\frac{\sum_i^N \xi_i \{\Theta_i, [R_i]_t\}_a}{[R]_{at}}}_{E4} \tag{23}$$





The right panel of Figure 8 shows the magnitude of terms from Eq.(23) for all events with the mean magnitude reported in the middle three rows of Table 2. Note that only $[\Theta]_a$ is shown in the figure since $[\Theta_1]_a$ and $[\Theta_2]_a$ are constants (see Table 2 for values). Generally speaking, runoff routing takes longer time with the increasing catchment drainage area for the two components as the elongation in flow path. By comparing the magnitudes from

the figure and table, it is obvious that the delay in catchment response is mainly contributed by the catchment-average runoff routing time (*E3*). The spatial covariance term (*E4*) is low, indicating that the contours of rainfall excess are not followed by the contours of isochrones for runoff routing (Woods & Sivapalan, 1999; Sangati, et al., 2009; Viglione, et al., 2010b; Volpi, et al., 2012). This is anticipated given the low elevation and mild slope topographic setups of the study region causing no orographic pattern in rainfall excess. Component-wisely speaking,

the subsurface routing is taking longer time than the surface one as shown in Figure 3. Under the relationship specified by Eq.(22), values of the total $E(\Theta)$ is in between the subsurface and surface $E(\Theta)$. We also observe from the figure that $E(\Theta)$ of the two components are getting closer to the total $E(\Theta)$ with the increase of the drainage area. This reflects the trend of change in mean $\xi$ where mean $\xi$ for the two components are getting closer from Swift to Tar given that the two $\psi$ values remain relatively unchanged. The June 2006 event is an example showing

that the subsurface process is characterized by negative spatial covariance. This is explained by its outlet concentrated cumulative rainfall (Figure 2). Moreover, by comparing $E(T)$, $E(\Theta)$, and $E(\Phi)$ in Table 2, we note that the delay in catchment response is increasing with drainage area; contribution to $E(\Phi)$ from the two stages are comparable in magnitude with $E(T)$ mostly larger than $E(\Theta)$.

### 4.3.2. Variance of Catchment Response Time

In the two-stage analytical framework, the variance of catchment response time is contributed by the variances introduced from the holding time of each of the stages and the covariance between holding time of the two stages. We write:

$$var(\Phi) = \underbrace{var(T)}_{Stage1} + \underbrace{var(\Theta)}_{Stage2} + \underbrace{2cov(T,\Theta)}_{Movement} \tag{24}$$

For stage 1, the variance of delay in rainfall excess generation for a rainfall excess component is provided as:

$$var_i(T) = \frac{|T_P|^2}{12} + \frac{\{T^2,[R_i]_a\}_t - |T_P|\{T,[R_i]_a\}_t - \frac{(\{T,[R_i]_a\}_t)^2}{[R_i]_{at}}}{[R_i]_{at}} \tag{25}$$

$var_i(T)$ represents the variance of instantaneous time with respect to the temporal distribution of rainfall excess; the

second term takes null for temporal uniform rainfall excess or rainfall excess concentrated purely on the event mid-point.

For total rainfall excess, the variance of delay in rainfall excess generation, $var(T)$, is correlated with $var_i(T)$ as (Appendix III):

$$var(T) = \sum_i^N \psi_i var_i(T) + \sum_i^N \psi_i [E_i(T) - E(T)]^2 \tag{26}$$

The first term is clearly the expectation of variance from all the other components. It signifies that the larger the

rainfall excess component, the stronger the control in dispersion of the total rainfall excess. The second term is the variability of $E_i(T)$ that arises since variance is not a linear operator. It measures the mean difference in the temporal



mass center between components to the total rainfall excess. We name the second term as *LT* because it may be interpreted as the squared sum of "time lag" (between each component to the total) in rainfall excess generation. The first and the second term account for the intra- and inter-component variability. Substituting in Eqs.(14), (16), & (25) to Eq.(26), a complete form is given as:

$$var(T) = \underbrace{\frac{|T_P|^2}{12}}_{v1} + \underbrace{\frac{\{T^2, \sum_i^N [R_i]_a\}_t - |T_P|\{T, \sum_i^N [R_i]_a\}_t - \frac{\sum_i^N (\{T, [R_i]_a\}_t)^2}{[R_i]_{at}}}{[R]_{at}}}_{v2} \qquad (27)$$
$$+ \underbrace{\frac{1}{[R]_{at}} \sum_i^N \frac{\left(\{T, [R_i]_a\}_t - \{T, \sum_i^N [R_i]_a\}_t\right)^2}{[R_i]_{at}}}_{LT}$$

5 Eq.(27) is a general form of Eq.(20) in V2010 with again the consideration of more than one flow component.

Results for Eq.(27) are illustrated in the left panel of Figure 9 and the first four rows of Table 3. Major source of *var(T)* is the variance of event duration (*v1*). However, the additional variance caused by the temporal interaction between rainfall excess and time (*v2*) is not negligible. This states that the distributions of rainfall excess of the events are not uniform in time (Woods & Sivapalan, 1999; Viglione, et al., 2010b). Additionally, event time series 10 of the two rainfall excess components are equally dispersed during the event period given the fairly close *var(T)* values (only the Swift case shows medium difference). This is exemplified by Figure 4 where the shapes of time series for the two components are quite close in the Fishing and Tar case but a bit more deviated in the Swift. Results from the figure and table also suggest that the magnitude of "time lag" term (*LT*) is irrelevant. A better visualization of reason is provided by the sample events time series in Figure 4. Most of the temporal variability of 15 rainfall is preserved in the time series as we can see from the shapes of *[P]ₐ[W₁]ₐ* and *[P]ₐ[W₂]ₐ*. Inspection on *var(T)* reveals that although the magnitudes of *v1*, *v2* and *LT* show no scale-dependency, their combination, *var(T)*, is increasing with drainage area.

For the runoff routing stage, we derive the variance of runoff routing time for any rainfall excess component, *varᵢ(Θᵢ)*, as:

$$var_i(\Theta_i) = \{\Theta_i\}_a + \frac{\{\Theta_i^2, [R_i]_t\}_a - 2[\Theta_i]_a\{\Theta_i, [R_i]_t\}_a - \frac{(\{\Theta_i, [R_i]_t\}_a)^2}{[R_i]_{at}}}{[R_i]_{at}} \qquad (28)$$

20 This variance again does not correspond to a single equation in V2010. The first term is the spatial variance of the runoff routing time. The second one accounts for the additional variance introduced by the interaction between time average rainfall excess and the runoff routing time. If the rainfall excess is spatially uniform or concentrated on the isochrones representing the mean runoff routing time (i.e. *[Θ]ₐ*), the second term vanishes.

We may derive the variance of delay in runoff routing, *var(Θ)*, as (see Appendix IV):

$$var(\Theta) = \sum_i^N \psi_i var_i(\xi_i \Theta_i) + \sum_i^N \psi_i [E_i(\xi_i \Theta_i) - E(\Theta)]^2 \qquad (29)$$

25 Similarly, *var(Θ)* has two terms accounting for the intra- and inter-component variability of runoff routing. The first term is a linear combination of *varᵢ(ξᵢΘᵢ)*; it highlights the combined effect from hydrology and geomorphology in the intra-component variability. The second term is the variance of *Eᵢ(ξᵢΘᵢ)*; it is again interpreted as the sum of "time lag" in runoff routing between rainfall excess components to the total. It quantifies the square of mean





distance in spatial mass center between all components to the total. The term $\xi_i$ is not a random variable related to space or time; thus, it can be moved out of the expectation or variance operator. Hence, $var(\Theta)$ may be rewritten as:

$$
\begin{aligned}
var(\Theta) = {} & \underbrace{\sum_i^N \psi_i \xi_i^2 \{\Theta_i\}_a}_{v3} \\
& + \underbrace{\frac{\sum_i^N \xi_i^2 \{\Theta_i^2, [R_i]_t\}_a - 2\sum_i^N \xi_i^2 [\Theta_i]_a \{\Theta_i, [R_i]_t\}_a - \frac{\sum_i^N \xi_i^2 (\{\Theta_i, [R_i]_t\}_a)^2}{[R_i]_{at}}}{[R]_{at}}}_{v4} \\
& + \underbrace{\sum_i^N \psi_i \left( \xi_i [\Theta_i]_a - \sum_i^N \xi_i \psi_i [\Theta_i]_a + \frac{\xi_i \{\Theta_i, [R_i]_t\}_a}{[R_i]_{at}} - \frac{\sum_i^N \xi_i \{\Theta_i, [R_i]_t\}_a}{[R]_{at}} \right)^2}_{L\Theta}
\end{aligned} \tag{30}
$$

The magnitudes of terms in Eq.(30) are plotted in the middle panel of Figure 9 with mean statistics listed in Table 3 (the middle four rows). Results suggest that $v3$ is the main contributor of $var(\Theta)$ compared to the additional spatial variance ($v4$). $v4$ is positively skewed as shown in the figure with negative mean, indicating that the event rainfall excess tends to be concentrated by the catchment (i.e. spatially unimodal pattern) (Zoccatelli, et al., 2011; Mei, et al., 2014). $v4$ is low in magnitude because, again, there is little spatial correlation between the location of isochrones for runoff routing and the rainfall excess under the study area's topographic setups. Component-wised comparison reveals that the variance of delay in runoff routing of the surface rainfall excess are smaller than the subsurface one. This is ascribed to the larger magnitude of $\Theta_1$ and $[R_1]_{at}$ than $\Theta_2$ and $[R_2]_{at}$. Besides, results suggest negligible "time lag" term ($L\Theta$) in contribution to the total variance of runoff routing, meaning that the spatial mass center of rainfall excess for the two rainfall excess components are fairly close to the total one. This is an expected result because of the highly similar spatial pattern of rainfall excess and runoff routing for the two components. Observations of the two sample events demonstrate that $v4$ for the November 2009 event is closer to null. This is substantiated by the generally more uniformly distributed rainfall excess pattern of the November event ($[R]_t$ in Figure 5). Moreover, we compare the values of $var(\Theta)$ and $var(T)$ from the results. Obviously $var(T)$ dominates $var(\Phi)$ where the mean of $var(T)$ are at least more than 3 times of the mean of $var(\Theta)$. Values of $var(\Phi)$ are increasing with the basin drainage areas due to the increases of both $var(T)$ and $var(\Theta)$.

The covariance term in Eq.(17) is often interpreted as an indicator of "movement of storm", resulting from the relaxation of "stationary rainfall" assumption. The so-called "movement of storm" is not just the geographic movement, it also accounts for the change in space-time dynamic of rainfall excess with respect to the runoff routing during the storm period (Viglione, et al., 2010a; Zoccatelli, et al., 2011; Nikolopoulos, et al., 2013; Mei, et al., 2014). The form of $cov_i(T,\Theta_i)$ is written as:

$$
cov_i(T, \Theta_i) = \frac{\{T, \{\Theta_i, R_i\}_a\}_t}{[R_i]_{at}} - \frac{\{T, [R_i]_a\}_t \{\Theta_i, [R_i]_t\}_a}{[R_i]_{at}^2} \tag{31}
$$

This term is the additional variance generated from the correlation in runoff generation and routing. Positive and negative covariance are interpreted as the centroid of rainfall excess moving towards the catchment portion with longer or shorter runoff routing time (near periphery or outlet) as the event evolves.

The covariance term in our multi-component assumption may be written as (see Appendix V):





$$cov(T,\Theta) = \sum_i^N \psi_i cov_i(T,\xi_i\Theta_i) + \sum_i^N \psi_i[E_i(T)E_i(\xi_i\Theta_i) - E(T)E(\Theta)] \qquad (32)$$

The covariance operator also results in two terms where the first one is the component-wise expectation of covariance between $T$ and $\xi_i\Theta_i$ with respect to $\psi_i$; it measures the coevolution of all rainfall excess components over catchment and event period. The second term is the covariance between $E_i(T)$ and $E_i(\xi_i\Theta_i)$; positive or negative value of the second term implies that rainfall excess components with temporal mass centers distance from the early phase of event are located closer to the catchment portion with larger or smaller routing time (catchment periphery or outlet). Based on this interpretation, the inter-component covariance should be very small in most of the cases happening in the nature; this is because there is no restriction that a rainfall excess component with time center further away from the event mid-point should be centered over isochrones with longer routing time or vice versa. $\xi_i$ is not subjected to the covariance operator and can be moved out. Combining Eqs.(31) & (32), $cov(T,\Theta)$ may be further written as:

$$cov(T,\Theta) = \underbrace{\sum_i^N \frac{\xi_i\{T,\{\Theta_i,R_i\}_a\}_t - \dfrac{\xi_i\{T,[R_i]_a\}_t\{\Theta_i,[R_i]_t\}_a}{[R_i]_{at}}}{[R]_{at}}}_{c} \qquad (33)$$
$$+ \underbrace{\sum_i^N \psi_i[\xi_i E_i(T)E_i(\Theta_i) - E(T)E(\Theta)]}_{LT\Theta}$$

Magnitudes of terms in Eq.(33) for the surface and subsurface component and the total are rendered in the right panel of Figure 9 with mean of terms reported in Table 3. Note that the magnitudes of terms have been multiplied by 2 given the mass conservation in Eq.(24). Values of the covariance terms are almost symmetrically distributed at 0 and slightly positively skewed. This observation indicates that there is no clear tendency for the storm movement. This is again explained by the fact that there is no preferred spatial pattern of rainfall over the study region with negligible orographic enhancement. The $LT\Theta$ term reveals an insignificant effect from the inter-component covariance between the temporal and spatial mass center of rainfall excess. This result supports our first guess on the magnitude of $LT\Theta$. Due to the low $LT\Theta$, $cov(T,\Theta)$ is mainly manipulated by $c$. Inspection on magnitudes of the two rainfall excess components demonstrates that the correlation between $T$ and $\Theta$ for the subsurface one is higher. Meanwhile, we observe an increase of $cov(T,\Theta)$ magnitude from the Swift to the Tar catchment, consisting of the positive scale dependency in magnitude of storm movement concluded in Mei et al. (2014) over the same area. In all, the movement effect of rainfall excess in variance of catchment response is relatively insignificant in the study region.

## 5.  Role of the Analytical Framework on Flood Characteristics

The rainfall and catchment surface properties are intimately related with the generation of flood. Specifically, the analytical framework quantities, $[R]_{at}$, $E(\Phi)$, and $var(\Phi)$, are correlated with the cumulative volume ($V$), centroid ($C$), and spreadness ($S$) of event flow time series, respectively (Sangati, et al., 2009; Viglione, et al., 2010a; 2010b; Mejía & Moglen, 2010; Volpi, et al., 2012). To address the question of how sensitive the framework quantities are to the flood characteristics, we propose sensitivity tests in this section. The $V$, $C$, and $S$, which quantify the catchment flood response are defined as:

$$V = \int_{T_F} Q(t)\,dt \qquad (34)$$



$$ C = \frac{\int_{T_F} t \cdot Q(t)\, dt}{\int_{T_F} Q(t)\, dt} \tag{35} $$

$$ S = \sqrt{\frac{\int_{T_F} (t - C)^2 Q(t)\, dt}{\int_{T_F} Q(t)\, dt}} \tag{36} $$

where $Q(t)$ is the simulated event flow time series; $T_F$ and $|T_F|$ correspond to the flood event period and its duration. $V$ reflects the magnitude of cumulative flow of a flood event while $C$ and $S$ are related to the shape of flood event hydrograph. Specifically, $C$ is the temporal location of mass center of the hydrograph which can be used to surrogate the time to peak (for single peak hydrographs); $S$ represents the temporal degree of dispersion with respect to $C$; typically for a unimodal event the larger $S$ indicates less than a peak for the hydrograph.

Results of the sensitivity tests are illustrated in Figure 10 and Table 4. Overall, the catchment-average cumulative rainfall excess ($[R]_{at}/TP/$) shows relatively high consistency with the cumulative flow volume, especially for the Fishing and Tar catchments where the mean of mean error ($ME$) are within 1 mm for the events. For the Swift cases, a fairly slight overestimation of $V$ by merely 3 mm (in terms of mean $ME$) is observed. Table 4 also provides the centered root mean square ($CRMS$) as an indicator of the random error in estimating $V$. Magnitudes of $CRMS$ are fairly small at around 1.5 mm, considering that these are produced based on cumulative volume. A comparison between $ME$ and $CRMS$ gives more insights on the performance of the analytical framework. Random error is the main error source for the Fishing and Tar cases, while in the Swift, systematic overestimation is more dominated. In all, the analytical framework provides reliable estimation on the cumulative volume, given the low magnitudes of $ME$ and $CRMS$.

Middle panel of Figure 10 demonstrates the correlation between expectation of catchment response time and centroid of flood event from the simulation. $E(\Phi)$ is positively correlated with $C$ but with obvious underestimation from the figure. The systematic underestimation on event centroid is about 35 hours for events from the smallest basin and increases to 40 hours for events from the largest one. The random components of error are within 20% of the systematic one in magnitude. This signifies that the main issue in estimation of $C$ is the systematic underestimation from the analytical framework. This underestimation lies in the simplified structure of the analytical framework compared to a distributed hydrologic model in both land surface and routing processes. In the land surface process during the early phase of the event, precipitation is principally used to fill the water capacity of catchment under the infiltration excess; after a certain time period, flow rate rises rapidly with the existence of precipitation because of the saturation excess process. This can be visualized by the sample events time series in Figure 2. Consequently, the inclusion of precipitations before the functioning of saturation excess advances the temporal mass center of rainfall, leading to underestimation of the mass center location. Furthermore, in the analytical framework a water parcel is approximated traveling at a constant speed once it enters the basin while the linear reservoir routing scheme of CREST only discharges a portion of the water amount from the total storage in a given grid cell, which in turn, increases the equivalent travelling time.



The spreadness is compared to the standard deviation of the catchment response time (square root of variance of the catchment response time) in the last panel of Figure 10. Systematic underestimation is still the major source of error in the estimation. Its magnitudes are about -12 hours and the magnitudes decrease with increase in spreadness. The underestimation in $S$ is also originated from the differences in structure of a distributed hydrologic model and the framework. During the early phase of event, the infiltration excess is the dominant mechanism for runoff generation. Under such a condition, the flow rate rises gradually and the hydrograph tends to be smooth, implying high spreadness. One can take the November 2009 event as an example; instead of having one rapid rising, the Swift catchment hydrograph has two rising limbs due to the switch in rainfall excess generation mechanism. And this bi-modal shape introduces larger spreadness compared to a unimodal rising shape. A similar argument regarding the influence of infiltration excess on runoff generation has been reviewed in Mei et al. (2014). Their study argues that the low sensitivity between shape error of rainfall and simulated runoff shown for the events is because most of the events from the Tar region do not have a bank-full condition. On the other hand, the equivalent travel time in the runoff routing of CREST is underestimated, and this underestimation is increasing with the length of water path given that $\Theta$ represents a cumulative sum along water path (Eq.(3)). This leads to underestimation in the variability of travelling time of CREST by the framework. To sum up, the analytical framework works better in predicting the spreadness of hydrograph compared to the centroid. Mean values of $ME$ for $S$ are approximately 34% of the $C$ case. Meanwhile, we observe relatively low values of the mean $CRMS$ for both the $S$ and $C$ estimation for the catchments (around 6 hours).

## 6. Conclusions

In this study we propose a hydrologic analytical framework based on a set of variables in catchment flood response (rainfall, runoff coefficient and runoff routing time). This framework is an expansion of V2010 under the consideration of multiple components in catchment flood response. As a first demonstration, the number of components is fixed at two, representing the surface and subsurface process simulated by the CREST model. The necessary framework parameters and event flow hydrographs are either outputs or estimates based on the results from CREST. We demonstrate the framework by a large number of flood events which occurred during 2003 to 2012 over the three catchments from the Tar River basin. Two of the events are selected as pilot events to provide detailed demonstrations of the framework. Sensitivity tests are rendered at last to investigate the correlation between framework and flood characteristics. Overall, the findings can be summarized as follow.

For the aspect of rainfall excess generation, we showed that the amount of rainfall excess generation was inverse proportional to catchment size. The most significant contribution came from the product term between space-time aggregated rainfall and runoff coefficients, while spatial and temporal correlation and movement effects were not significant. In addition, it was shown that the subsurface component outperformed the surface component of runoff in the contribution to rainfall excess generation, but this difference diminished in larger catchments.

The expectation of catchment response time was also investigated. We found that the total rainfall excess generation time is a linear combination of expected generation time from the other components with respect to the rainfall excess ratio. The total runoff routing time is a combination of routing time from the others related to the rainfall excess ratio and the coefficient accounts for the hydrologic and geomorphologic effects. From the results we conclude that both of the rainfall excess generation and the runoff routing stage are important to the delay in the response of a catchment. The length of the rainfall event and the magnitude of the runoff routing time field play a significant role in controlling the timing of the hydrograph. Delay in response due to the spatial and temporal





correlation term is low. The total catchment response time was shown to be closer to the subsurface rainfall excess one, indicating a higher degree of influence, which agrees with the higher rainfall excess ratio for the subsurface component. But the value gap between components is mitigating from small to large catchment area.

For the variance of catchment response time, our findings showed that the total variance in rainfall excess generation comes from two parts—the linear combination of all components variance and the variance of expected rainfall excess generation time for components. These two parts account for the intra- and inter-component variability, respectively. The total variance in delay due to the runoff routing stage is consisted of two parts—a combination of variance from all the other components and the variance of expected runoff routing time with the participation of the hydrologic and geomorphologic related coefficient. Analogously, the covariance between holding time of the two stages is also consisted of two parts—they are the expectation of component covariance and the covariance between expectations of rainfall excess generation and runoff routing. Results revealed that variance of the rainfall excess generation stage is of higher importance than that of the runoff routing stage. For stage 1, the variance from rainfall duration was more important than the additional variance from temporal interactions between rainfall excess and time. For stage 2, the spatial variance of runoff routing time outperformed the additional variance that arose from the spatial interaction between rainfall excess and runoff routing. Additionally, variance of the surface component was closer to the total variance, indicating a higher degree of influence. Furthermore, the inter-component variability was negligible compared to the intra-component variability.

Results from the sensitivity test revealed that the framework is characterized by relatively low random errors in estimating the flood characteristics. A slight overestimation was found in the Swift catchment on the estimation of cumulative flow volume. Systematic underestimation in event centroid and spreadness were notable, especially for the timing issue, which shows increasing trend with the catchment scale. Moreover, the underestimation of spreadness was reduced with the increase in magnitude of spreadness.

From the herein analytical framework results, we showed that magnitudes of the new "time lag" terms are low. We believe this is not a general finding because the surface runoff coefficient was represented by a constant imperviousness ratio ($I_M$) for this study and the runoff routing times for the two components had the same spatial pattern (differed merely in magnitude by the constant $\alpha$). Future studies will need to replace the constant imperviousness ratio by a spatially distributed variable to mimic the spatial variability of the very fast flood response of a catchment. Also, we suggest the use of a spatially varied $\alpha_i$ to better represent the differences in routing for components. This is particularly useful in analyzing the flood response of urbanized catchments where the differences in runoff generation and routing are quite obvious between the highly impervious urban surface (e.g. roads, rooftops, parking lots, etc.) and pervious suburban or rural surface (Smith, et al., 2002; Mejía & Moglen, 2010; Mejía, et al., 2015). We believe the "time lag" terms could be important for flood response of the urbanized catchment and our new framework can serve as a diagnostic tool to verify the significance of these terms.

We acknowledge certain limitations of our analytical framework study. The framework variables and flow simulation are dependent on the hydrologic model devised in this study (e.g. imperviousness areas, coefficient $\alpha$, etc.). Since the retrievals of framework variables are based on the model structure and parameterization, the way a variable is calculated could vary across models, while in certain models such an explicit parameter may not be available. An alternative path to circumventing this issue is to apply directly observed data for the calculation of the analytical framework variables. For instance, the spatial patterns of runoff coefficients are provided in certain



data rich locations (Merz, et al., 2006; Dhakal, et al., 2012), while at global scale, runoff coefficients may be retrieved from satellite-derived soil moisture fields (Penna, et al., 2011; Massari, et al., 2014).

*Acknowledgements:* The current study was supported by a research grant from the Connecticut Institute for Resilience and Climate Adaptation.





**Appendix**

I. Expectation of rainfall excess generation time

To calculate the expectation of rainfall excess generation time, follow the definition of expectation and write

$$E(T) = \frac{1}{|T_P|} \int_{T_P} \frac{1}{|A|} \int_A (t \cdot f_R)\, da\, dt = \frac{1}{|T_P|} \int_{T_P} \frac{1}{|A|} \int_A \left( t \sum_i^N \psi_i f_{R_i} \right) da\, dt$$

$$= \sum_i^N \psi_i \frac{1}{|T_P|} \int_{T_P} \frac{1}{|A|} \int_A (t \cdot f_{R_i})\, da\, dt$$

Note that by definition $E_i(T)$ is

$$E_i(T) = \frac{1}{|T_P|} \int_{T_P} \frac{1}{|A|} \int_A (t \cdot f_{R_i})\, da\, dt$$

Hence

$$E(T) = \sum_i^N \psi_i E_i(T)$$

II. Expectation of runoff routing time

For the expectation of runoff routing time, follow the definition of expectation and write

$$E(\Theta) = \frac{1}{|T_P|} \int_{T_P} \frac{1}{|A|} \int_A (\theta \cdot f_R)\, da\, dt = \frac{1}{|T_P|} \int_{T_P} \frac{1}{|A|} \int_A \left( \theta \sum_i^N \psi_i f_{R_i} \right) da\, dt$$

$$= \frac{1}{|T_P|} \int_{T_P} \frac{1}{|A|} \int_A \left( \sum_i^N \xi_i \theta_i \psi_i f_{R_i} \right) da\, dt = \sum_i^N \psi_i \xi_i \frac{1}{|T_P|} \int_{T_P} \frac{1}{|A|} \int_A (\theta_i f_{R_i})\, da\, dt$$

By definition we have

$$E_i(\Theta_i) = \frac{1}{|T_P|} \int_{T_P} \frac{1}{|A|} \int_A (\theta_i f_{R_i})\, da\, dt$$

So, $E(\Theta)$ may be derived as:

$$E(\Theta) = \sum_i^N \psi_i E_i(\xi_i \Theta_i)$$

III. Variance of rainfall excess generation time

For the variance of rainfall excess generation time, we first calculate $E(T^2)$. Following the same method in $E(T)$, we have

$$E(T^2) = \sum_i^N \psi_i E_i(T^2)$$

Thus

$$var(T) = E(T^2) - [E(T)]^2 = \sum_i^N \psi_i E_i(T^2) - [E(T)]^2$$




$$= \sum_i^N \psi_i E_i(T^2) - \sum_i^N \psi_i [E_i(T)]^2 + \sum_i^N \psi_i [E_i(T)]^2 - 2[E(T)]^2 + [E(T)]^2$$

$$= \sum_i^N \psi_i \{E_i(T^2) - [E_i(T)]^2\} + \sum_i^N \psi_i \{[E_i(T)]^2 - 2E_i(T)E(T) + [E(T)]^2\}$$

$$= \sum_i^N \psi_i var_i(T) + \sum_i^N \psi_i [E_i(T) - E(T)]^2$$

The first term is the expectation of variance of $T$; the second term is the variance of expectation of $T$. These two terms account for the intra- and inter-component variability.

IV.  Variance of runoff routing time

Similarly, for the variance of runoff routing time, calculate $E_i(\Theta_i^2)$ as the first step,

$$E(\Theta^2) = \sum_i^N \psi_i E_i(\xi_i^2 \Theta_i^2)$$

Write (treat the product of $\xi_i$ and $\Theta_i$ as one variable)

$$var(\Theta) = E(\Theta^2) - [E(\Theta)]^2 = \sum_i^N \psi_i E_i(\xi_i^2 \Theta_i^2) - 2[E(\Theta)]^2 + [E(\Theta)]^2$$

$$= \sum_i^N \psi_i E_i(\xi_i^2 \Theta_i^2) - \sum_i^N \psi_i [E_i(\xi_i \Theta_i)]^2 + \sum_i^N \psi_i [E_i(\xi_i \Theta_i)]^2 - 2[E(\Theta)]^2 + [E(\Theta)]^2$$

$$= \sum_i^N \psi_i \{E_i(\xi_i^2 \Theta_i^2) - [E_i(\xi_i \Theta_i)]^2\} + \sum_i^N \psi_i \{[E_i(\xi_i \Theta_i)]^2 - 2E_i(\xi_i \Theta_i)[E(\Theta)] + [E(\Theta)]^2\}$$

$$= \sum_i^N \psi_i var_i(\xi_i \Theta_i) + \sum_i^N \psi_i [E_i(\xi_i \Theta_i) - E(\Theta)]^2$$

Similar to $var(T)$, the expression has two terms accounting for the intra- and inter-component variability. The first term is the expectation of variance of $\xi_i \Theta_i$; the second one represents the variance of expectation of $\xi_i \Theta_i$.

V.  Covariance between rainfall excess generation and runoff routing time

For the covariance, it is trivial to show

$$E(T\Theta) = \sum_i^N \psi_i E_i(T \xi_i \Theta_i)$$

Write

$$cov(T,\Theta) = E(T\Theta) - E(T)E(\Theta) = \sum_i^N \psi_i E_i(T \xi_i \Theta_i) - E(T)E(\Theta)$$

$$= \sum_i^N \psi_i E_i(T \xi_i \Theta_i) - \sum_i^N \psi_i E_i(T)E_i(\xi_i \Theta_i) + \sum_i^N \psi_i E_i(T)E_i(\xi_i \Theta_i) - E(T)E(\Theta)$$

$$= \sum_i^N \psi_i cov_i(T, \xi_i \Theta_i) + \sum_i^N \psi_i [E_i(T)E_i(\xi_i \Theta_i) - E(T)E(\Theta)]$$

The first term is the expectation of covariance between $T$ and $\xi_i \Theta_i$. The second term is the covariance between expectation of $T$ and of $\xi_i \Theta_i$.





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





Table 1. Mean magnitudes of terms in Eq.(9) in mm/h.

| Term | Swift | | | Fishing | | | Tar | | |
|---|---|---|---|---|---|---|---|---|---|
| | Sub-surface | Surface | Total | Sub-surface | Surface | Total | Sub-surface | Surface | Total |
| R1 | 0.212 | 0.149 | **0.361** | 0.138 | 0.100 | **0.238** | 0.103 | 0.088 | **0.193** |
| R2 | -0.005 | \ | -0.005 | -0.001 | \ | -0.001 | 0.000 | \ | 0.000 |
| R3 | 0.011 | \ | 0.008 | 0.005 | \ | 0.005 | 0.004 | \ | 0.004 |
| R4 | 0.004 | \ | 0.005 | 0.001 | \ | 0.001 | 0.002 | \ | 0.002 |
| $[R]_{at}$ | *0.222* | 0.149 | 0.371 | *0.144* | 0.100 | 0.243 | *0.108* | 0.090 | 0.198 |
| | | | | | | | | | |
| MV | 0.005 | \ | 0.005 | 0.001 | \ | 0.001 | 0.002 | \ | 0.002 |

Table 2. Same as in Table 1 but for Eqs.(16) & (23).

| Term | Swift | | | Fishing | | | Tar | | |
|---|---|---|---|---|---|---|---|---|---|
| | Sub-surface | Surface | Total | Sub-surface | Surface | Total | Sub-surface | Surface | Total |
| E1 | 32 | 32 | 32 | 38 | 38 | 38 | 40 | 40 | 40 |
| E2 | -0.78 | -4.5 | -2.3 | -4.3 | -6.3 | -5.1 | -2.0 | -4.1 | -3.0 |
| E(T) | 32 | 28 | **30** | 33 | 31 | **32** | 38 | 36 | **37** |
| | | | | | | | | | |
| E3 | 30 | 9 | 21 | 33 | 20 | 27 | 37 | 25 | 32 |
| E4 | -1.2 | -0.22 | -0.64 | -1.0 | -0.37 | -0.68 | -2.6 | -1.2 | -1.8 |
| E($\Theta$) | 29 | 9.2 | 21 | 32 | 20 | 27 | 35 | 24 | 30 |
| | | | | | | | | | |
| $\psi$ | 0.57 | 0.43 | \ | 0.57 | 0.43 | \ | 0.54 | 0.46 | \ |
| $\xi$ | 0.70 | 2.27 | \ | 0.84 | 1.34 | \ | 0.85 | 1.27 | \ |
| | | | | | | | | | |
| E($\Phi$) | *61* | 37 | 51 | *65* | 51 | 59 | *72* | 59 | 66 |



Table 3. Same as in Table 1 but for Eqs.(27), (30) & (33).

| Term | Swift | | | Fishing | | | Tar | | |
|---|---|---|---|---|---|---|---|---|---|
| | Sub-surface | Surface | Total | Sub-surface | Surface | Total | Sub-surface | Surface | Total |
| v1 | 536 | 536 | 536 | 769 | 769 | 769 | 740 | 740 | 740 |
| v2 | 73 | 103 | 86 | 19 | 22 | 21 | 60 | 60 | 61 |
| LT | \ | \ | 7.6 | \ | \ | 2.4 | \ | \ | 2.5 |
| var(T) | 609 | 638 | **629** | 788 | 791 | **792** | 800 | 800 | **803** |
| | | | | | | | | | |
| v3 | 223 | 21 | 112 | 150 | 59 | 106 | 340 | 152 | 244 |
| v4 | -22 | -1.0 | -8.8 | -7.7 | -2.6 | -5.1 | -10 | -3.4 | -6.7 |
| LΘ | \ | \ | 0.65 | \ | \ | 0.18 | \ | \ | 0.58 |
| var(Θ) | 201 | 20 | 103 | 142 | 56 | 101 | 330 | 149 | 238 |
| | | | | | | | | | |
| c | 3.3 | -1.1 | -1.2 | -26 | -16 | -22 | -28 | -19 | -23 |
| LTΘ | \ | \ | -0.59 | \ | \ | -0.56 | \ | \ | -0.82 |
| cov(T,Θ) | 3.3 | -1.1 | -1.8 | -26 | -16 | -23 | -28 | -19 | -24 |
| | | | | | | | | | |
| var(Φ) | 813 | *658* | 730 | *904* | 831 | 871 | *1102* | 930 | 1017 |

Table 4. Statistics of the sensitivity tests.

| Basin | Mean Error | | | Centered Root Mean Square | | |
|---|---|---|---|---|---|---|
| | V (mm) | C (h) | S (h) | V (mm) | C (h) | S (h) |
| Swift | 2.9 | -35.2 | -12.7 | 1.6 | 7.6 | 5.7 |
| Fishing | -0.7 | -36.9 | -10.7 | 1.6 | 3.8 | 5.5 |
| Tar | 0.6 | -39.6 | -15.2 | 1.3 | 6.5 | 5.4 |





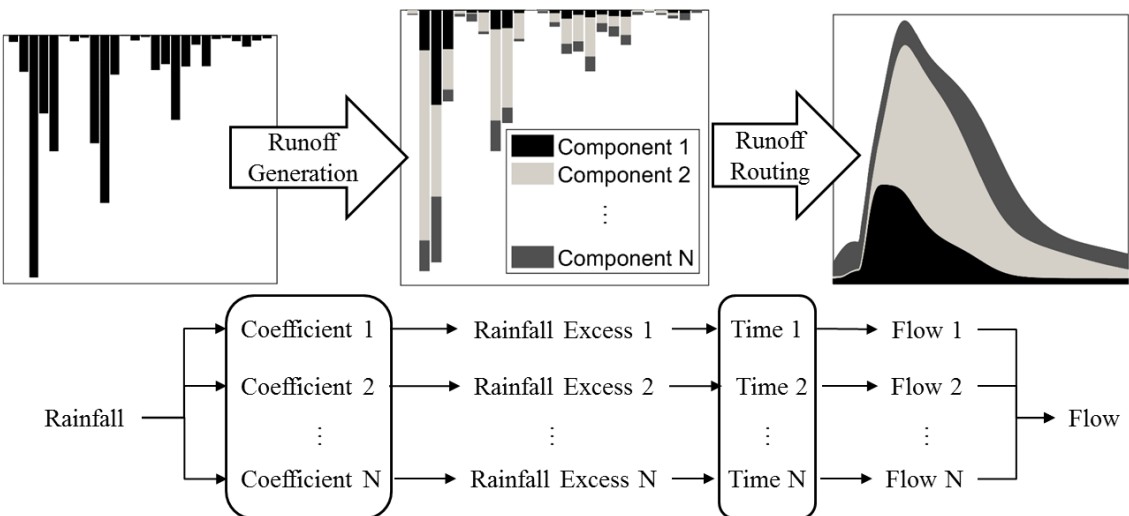

Figure 1. Schematic of the Analytical Framework.





Figure 2. Event rainfall map and time series of rainfall and runoff for the two representative events.



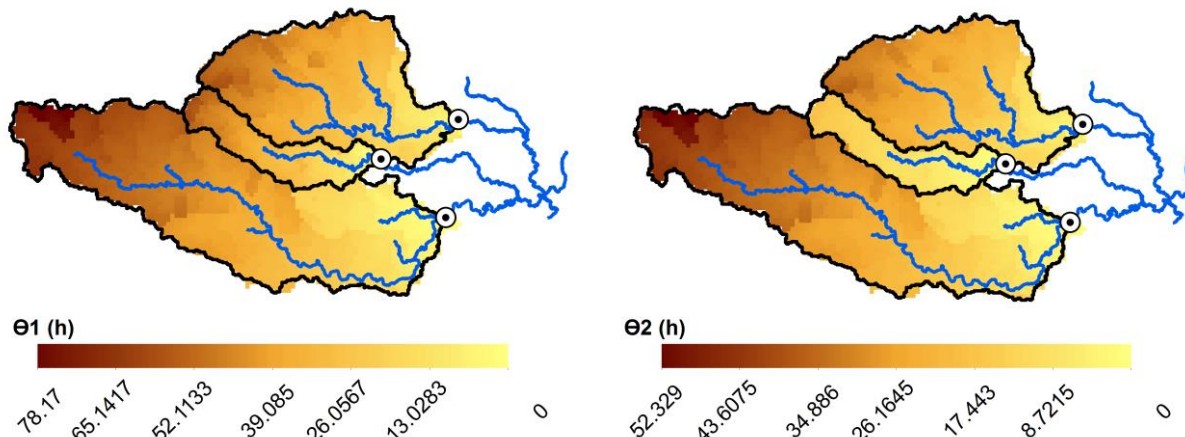

Figure 3. Runoff routing time for the study basins.





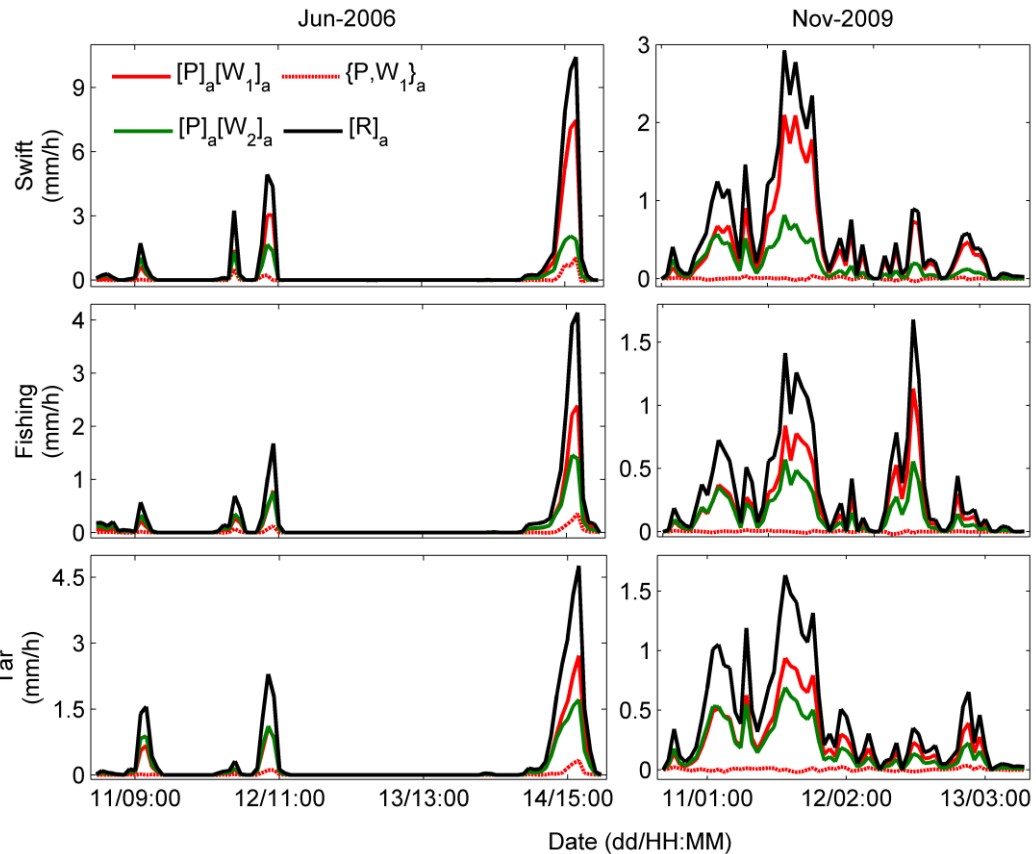

Figure 4. Time series showing the spatial averaged terms in Eq.(7) for the different rainfall excess components of the two representative events.









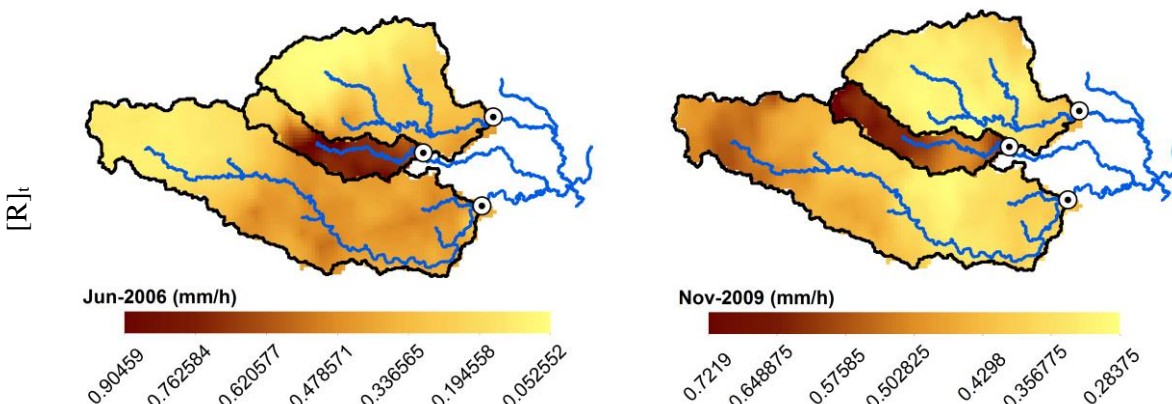

Figure 5. Same as Figure 4 but for temporal averaged terms in Eq.(8).





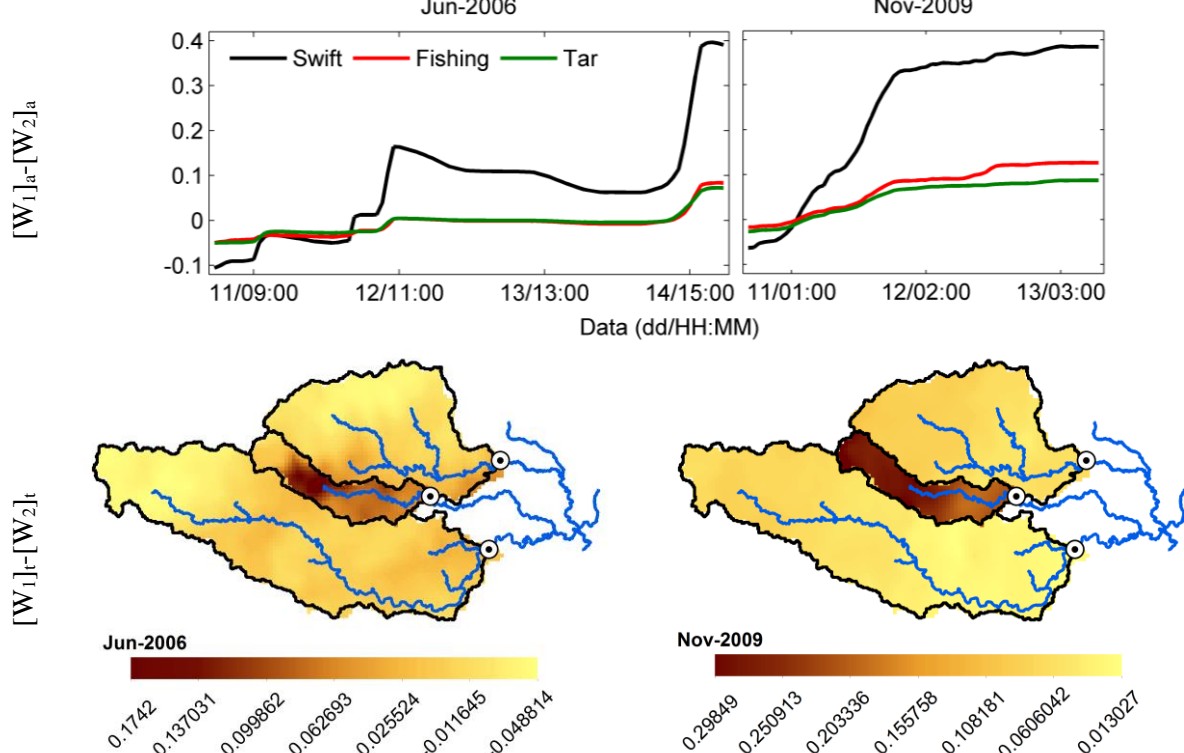

Figure 6. Differences between catchment-average and storm-average runoff coefficient.





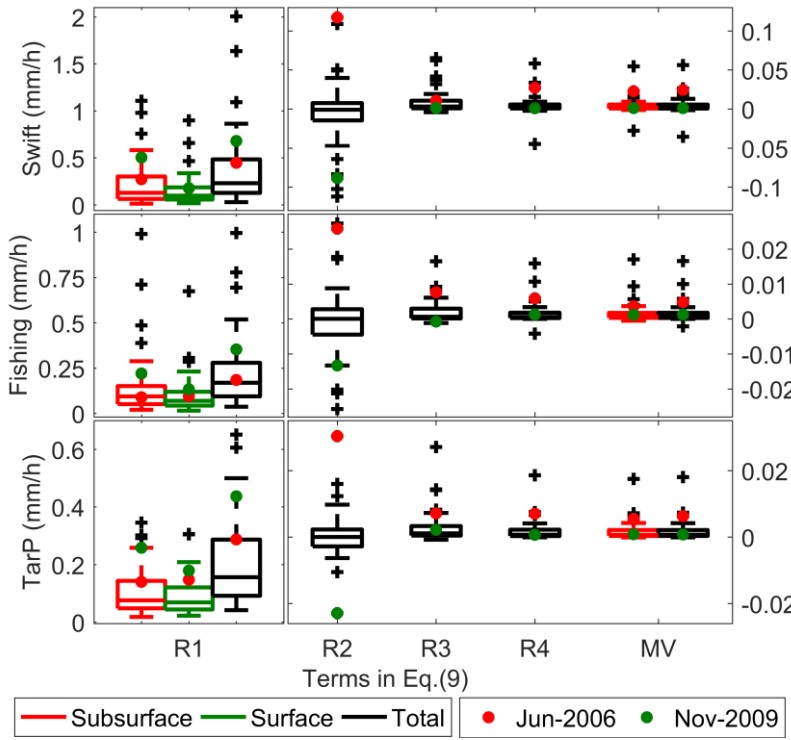

Figure 7. Boxplot showing the spatiotemporal averaged terms in Eq.(9) for all events from the study basins.





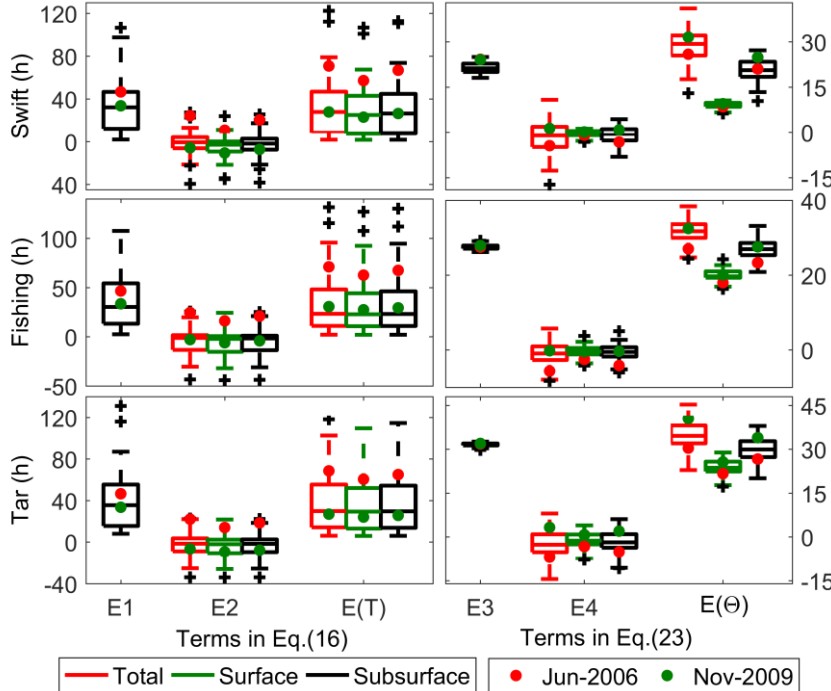

Figure 8. Same as Figure 7 but for Eqs.(16) & (23).





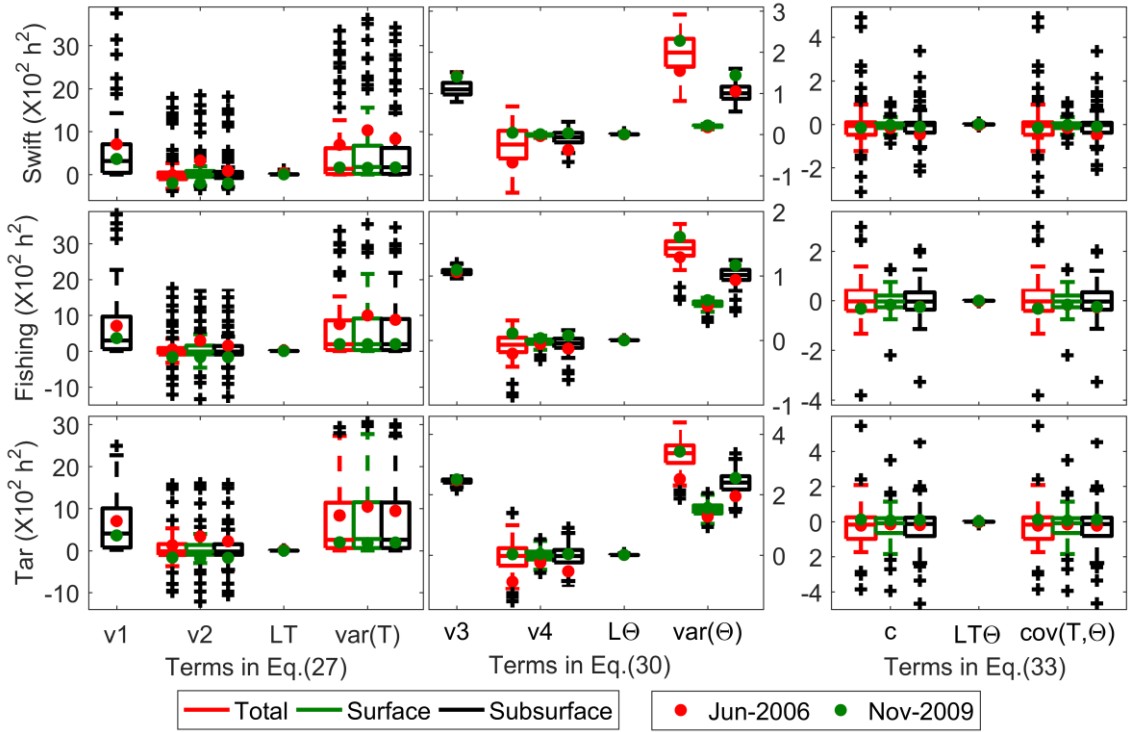

Figure 9. Same as Figure 7 but for Eqs.(27), (30) & (33).





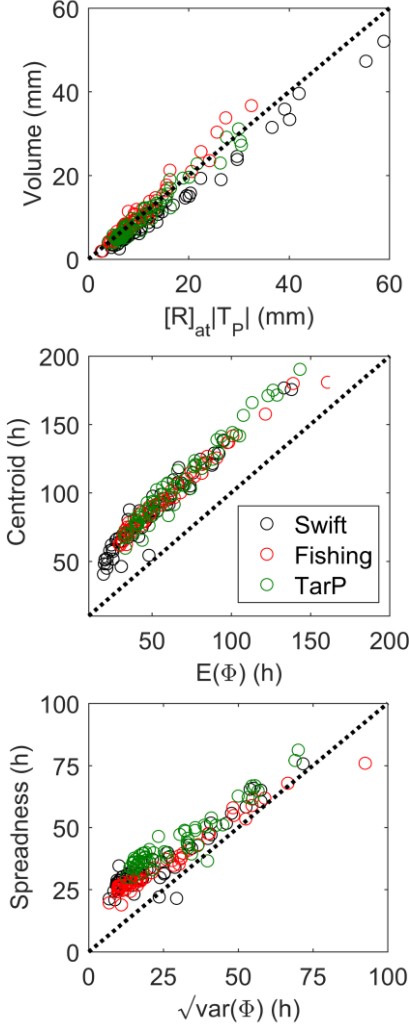

Figure 10. Sensitivity test of the analytical framework outputs to hydrograph properties.