# Peer review of "A Synthesis of Space-time Variability in Multi-Component Flood Response"

_Hydrology and Earth System Sciences, 2016_

## Referee Comment (RC1) · Anonymous Referee #1 · 15 Jul 2016

The paper provides a generalization of the analytical framework provided by Viglione et al. (2010), which in turns refers to Woods and Sivapalan (1999). With respect to Viglione et al. (2010) the authors introduce a multicomponents analysis of the flood streamflow but, on the other hand, in trating travel time-related issues they do not distinguish between hillslope and channel flow routing.

While the theoretical arguments and developments are interesting and well posed, I believe the paper should be reinforced with more attention to congruency with the hydrographs observed in real events.

In particular, in my opinion, more emphasys and major discussion should be provided about the comparison of the analytical results with the observed values of time to peak and hydrograph spreadness of the two observed events.

[Figure]

The weight and consequences of their assumption discarding the difference between hillslope and channel flow velocity should more deeply analyzed and discussed.

Results presented in Tables, if I correctly understood, are mean values referred to different observed events. Why not to show values referred to both the pilot events that look quite different for their distribution in time ?

Obviously, the analytical framework requires data from a model for its application, nevertheless results and comments would be more acceptable if they were used also for a diagnostic analysis of the model itself.
* * *

---

## Author Comment (AC1) · 27 Jul 2016

The authors would like to thank the Anonymous Referee #1 for his/her critical points. They are all very important aspects that helped improve the soundness and quality of the presentation of our results. Below is description of revisions addressing the Referee's comments. In the attachment we include marked-up document of the revised manuscript.

Specifically, there are four points in the Referee's comments. In point #1, the Referee asked for more emphasis and discussion on comparison of the analytical framework results with the observation. We therefore included additional scatter plots of hydrograph properties derived from the analytical framework with respect to the observation in the original Figure 10. Corresponding statistics have also been added in the original Table

4. This resulted in major revision of Section 5 and paragraph 5 of Section 6. Please refer to these sections, as well as Figure 10 and Table 4 of the revised manuscript.

In point #2, the Referee suggested that we analyze and discuss more deeply the reasons of lumping hillslope and channel flow velocity in our new framework. Here are our responses. First, it should be noted that the hillslope and channel routing processes are considered individually by CREST model and thus the runoff routing time $\Theta$ attained by the analytical framework should represent the sum of routing times of both processes. Besides, given that CREST uses the same parameterization to model these two routing times (they only differ in terms of parameter values), we see no additional benefit for the analytical framework, and rather an increase of framework complexity, to decompose runoff routing into hillslope and channel routing. To make our argument clearer, we added discussion in lines 27 to 30 on page 5 and lines 22 to 27 on page 6.

In point #3, the Referee suggested to show the values of terms for the two pilot events because they are characterized by different space and time patterns. Therefore, we added descriptions for the value of terms of the events and their implications in lines 6-7 of page 5, lines 22-24 of page 8, lines 9-10 of page 10, lines 19-20 of page 11, lines 18-20 of page 12 and lines 22-23 of page 13. The reason we did not add this information in the tables is because this will make the tables very long (three times the size of the original table). Also, since the values of these terms for the pilot events have already been shown in the figures, we feel that adding such information in text should be sufficient.

The main concern in point #4 is the hydrologic representativeness of the CREST simulation results, which in turn refers to the representativeness of the analytical framework. To articulate this point, we added content to describe the value of error metrics of the two pilot events in lines 5-7 of page 5. Also, we refer to the content provided in lines 32-34 of page 4 and lines 2-4 of page 5 for the performance of the simulations.

[Figure]

Please also note the supplement to this comment:
http://www.hydrol-earth-syst-sci-discuss.net/hess-2016-255/hess-2016-255-AC1-supplement.pdf

———————————————————

[Figure]

**Supplement:**

[revised manuscript text omitted]

Commented [MY21]: Referee #1, point #1

---

## Referee Comment (RC2) · Anonymous Referee #2 · 13 Oct 2016

This manuscript generalizes the analytical framework of V2010 to represent multi-component flood responses. To achieve this, the authors convert the catchment rainfall forcing into more than one rainfall excess component, according to various land cover types. These rainfall excess components are then subjected to different routing schemes associated with the different land cover classes. The output hydrograph is a combination of hydrographs from all components.

**Major comments**

- The introduction of the paper is complete and frames the problem interestingly. I have not additional comments there.

- The manuscript requires additional editorial work. Below I indicate some edits that I was able to identify, but the authors need to make sure that typos and grammatical mistakes are completely resolved.

- I'm confused about the number and the basis for choosing the multi-components of the flood response. In the introduction section, it seems the components are chosen based on different land use classes (see, e.g., lines 35-37, P2). Then, in lines 18-20, P5, it appears that only two components are considered and on the basis of flowpaths, namely surface or subsurface, rather than land cover classes. This requires clarification.

- In some instances, the manuscript refers the reader to equations in another published paper, i.e. V2010. This should be avoided to make the manuscript readable and self-contained. For example, this occurs in lines 3-5, P8, but in other places as well.

- The conclusion section could be organized using bullet points. Additionally, it should make clear (e.g., line 28. P16) that they are based on the selected study basins.

**Minor comments**

L36, P2: Should read "… excess component by the various …"

L42, P2: Should read "… the most dominant for a component …"

L16-17, P3: "The readers are encouraged to read Mei & Anagnostou (2015) for details on the hydrology of the study area", consider rephrasing to, e.g., "The reader is refer to Mei & Anagnostou (2015) for further details on the hydrology of the study area."

L24-25, P3: Rephase "Another meteorology data is the potential 25 evapotranspiration (PET) data available …"

L28-29, P3: Should read "… record available from the United States Geological Survey (USGS) …"

L32, P3: Should read "…are its parsimonious data requirement …"

L11, P4: It would help to include here the spatial resolution of the CREST model used in this study.

L23, P4: Should read "… parameter optimization algorithm adopted in CREST …"

L19-20, P6: Rephrase.

L32, P6: Remove or replace "… using the parsimonious rainfall-runoff equation …"
L18, P7: Rephrase "…in the mature and decade phase of the event …"

L20, P7: Rephrase "…the differences start at negative …"

L26, P7: Should read "… aggregated maps for terms in Eq. (8) …"

L9, P8: Should read "…The magnitudes of terms in Eq.(9) …"

L18, P8: Rephrase "…diminishing in magnitude with increase in scale is a result …" In the present form, this sentence is difficult to understand.

L21, P8: Should read "…the subsurface component generally outperforms the surface one …"

L3-4, P9: Not sure what the authors mean by "… a convex combination …", please explain or modify.

L10, P9: Should read "…where $|T_P|$ is the duration of the rainfall event."

L2, P10: Should this be "… by rendering negative the mean $E2$ …"? Otherwise, this needs further clarification.

L3-4, P11: Rephrase this sentence, in present form is confusing.

L5, P12: This is not needed and it is actually distracting.

L6, P12: Should read "… The major source of …"

L20, P12: Not needed.

L16, P15: Should read "The middle panel of Figure 10 …"

---

## Author Comment (AC2) · 19 Oct 2016

Hydrology and Earth System Science

Manuscript No.: hess-2016-255

Title: A Synthesis of Space-time Variability in Multi-Component Flood Response

Authors: Yiwen Mei, Xinyi Shen and Emmanouil N. Anagnostou

**NOTE:**

Reviewers' original comments are in 10pt Courier New type with yellow highlighting.

*Text from the revised manuscript is in bold italic font.*

Page/Line/Figure numbers given by reviewers were left unchanged in their text and refer to the originally submitted version of the manuscript

**Major comments**

1. The introduction of the paper is complete and frames the problem interestingly. I have not additional comments there.

Thank you.

2. The manuscript requires additional editorial work. Below I indicate some edits that I was able to identify, but the authors need to make sure that typos and grammatical mistakes are completely resolved.

Thank you. We have further edited the manuscript.

3. I'm confused about the number and the basis for choosing the multi-components of the flood response. In the introduction section, it seems the components are chosen based on different land use classes (see, e.g., lines 35-37, P2). Then, in lines 18-20, P5, it appears that only two components are considered and on the basis of flow paths, namely surface or subsurface, rather than land cover classes. This requires clarification.

This is a very good point and it definitely needs to be clarified. The separation of rainfall excess into components is based on the heterogeneity of vertical soil layers. Each of the sub-surface layers is associated with a flow path. This is typically represented in distributed hydrologic models implicitly using the multiple reservoirs approach, or in more advanced models through explicit equations. Land cover types are then used to characterize the distribution of the excess rainfall over the vertical soil layers. For example,

for an urbanized land cover a large portion of the excess rainfall is located within the very shallow layer (the first layer) of the soil while there is a shift of the excess rain water from the shallow to the deeper layers for more vegetated land cover types. To articulate these aspects, we have applied several modifications throughout the manuscript. In P1 line 2-4 of the abstract, we revised as:

***Catchment flood response consists of multiple components of flow originating from different surface and sub-surface layers. This study proposes an extension of Viglione et al. (2010a) analytical framework to represent the dependence of catchment flood response to the different runoff generation processes.***

In P2 line 28-32, we revised as:

***In this sense, the different runoff generation processes associated with vertical heterogeneous catchment layers are lumped together into a single flood response (Woods & Sivapalan, 1999; Viglione, et al., 2010b). Numerous experimental studies, though, have demonstrated that catchment flood response can be identified as multiple components originating from different catchment layers and associated with different flow paths (Weiler, et al., 2003; Liu, et al., 2004; Gonzales, et al., 2009).***

In P2 line 36-38, we revised as:

***Catchment rainfall forcing is converted to more than one rainfall excess components associated with various surface and subsurface layers. These rainfall excess components are subjected to different flow paths and routing schemes.***

In P18 line 25-28, we revised as:

***This is particularly useful in analyzing the flood response of urbanized catchments where the distribution of excess rainfall into different vertical soil layers is quite different between the highly impervious urban areas (e.g. roads, rooftops, parking lots, etc.) and the more pervious suburban or rural areas of the basin (Smith, et al., 2002; Mejía & Moglen, 2010; Mejía, et al., 2015).***

*4. In some instances, the manuscript refers the reader to equations in another published paper, i.e. V2010. This should be avoided to make the manuscript readable and self-contained. For example, this occurs in lines 3-5, P8, but in other places as well.*

Thank you, we see the point here. We provided the meaning of each term instead of referring the readers to V2010. There are five modifications reflected in the manuscript.

In P8 line 14-17:

***Term R1 represents the sum of product between the catchment-average storm rainfall and runoff coefficient for all components. Term R2/R3 is the sum of temporal/spatial covariance between the catchment-/storm-average rainfall and runoff coefficient. R4 is the sum of temporal covariance between spatial variation of precipitation and runoff coefficient.***

In P9 line 25-26:

*E1 refers to the event half-duration and E2 is the expectation of time distance from the event midpoint to the temporal mass center of catchment-average rainfall excess.*

In P11 line 5-6:

*E3 stands for the spatial mean of runoff routing time in the hillslope and channel network. E4 is the expected distance from the geomorphologic center of catchment to the centroid of storm-average rainfall excess.*

In P12 line 12-15:

*Term v1 stands for the variance in time generated by a temporal invariant catchment-average rainfall excess. Term v2 represents component-wised mean of additional variance caused by the temporal variation in catchment-average rainfall excess. The last term is named LT and represents the mean square of "time lag" (between each component to the total) in rainfall excess generation.*

In P13 line 16-18:

*Term v3 represents the variance in time generated by a spatial invariant storm-average rainfall excess. Term v4 is the mean of additional variance caused by the spatial variation in storm-average rainfall excess. The term LΘ represents the mean of "time lag" in runoff routing between rainfall excess components to the total.*

5. The conclusion section could be organized using bullet points. Additionally, it should make clear (e.g., line 28. P16) that they are based on the selected study basins.

Thanks for the suggestion but we choose to go with paragraph instead of bullet points. In addition, the sentence in P17 line 21-22 has been rephrased as:

*The findings from this study are summarized below.*

**Minor comments**

1. L36, P2: Should read "… excess component by the various …"

The whole sentence in P2 line 36-37 has been changed as:

*Catchment rainfall forcing is converted to more than one rainfall excess components associated with various surface and subsurface layers.*

2. L42, P2: Should read "… the most dominant for a component …"

Done (P3 line 1).

*3. L16-17, P3: "The readers are encouraged to read Mei & Anagnostou (2015) for details on the hydrology of the study area", consider rephrasing to, e.g., "The reader is refer to Mei & Anagnostou (2015) for further details on the hydrology of the study area."*

The sentence (P3 line 17-18) has been modified as:

***The reader is referred to Mei & Anagnostou (2015) and Mei et al. (2014) for details on the hydrology of the study area.***

*4. L24-25, P3: Rephrase "Another meteorology data is the potential evapotranspiration (PET) data available …"*

The sentence (P3 line 25-27) has been rephrased as:

***Another atmospheric forcing dataset used in this study is the potential evapotranspiration (PET) available from the North American Regional Reanalysis (NARR) at 3-hourly and 32 km resolution (Mesinger et al., 2006).***

*5. L28-29, P3: Should read "… record available from the United States Geological Survey (USGS) …"*

Done (P3 line 29-30).

*6. L32, P3: Should read "…are its parsimonious data requirement …"*

Done (P3 line 33).

*7. L11, P4: It would help to include here the spatial resolution of the CREST model used in this study.*

We reported the model grid domain resolution in P4 line 26 instead.

***…the model is set up over the three study catchments with 1 km spatial resolution…***

*8. L23, P4: Should read "… parameter optimization algorithm adopted in CREST …"*

Done (P4 line 23).

*9. L19-20, P6: Rephrase.*

The sentence on P6 line 24-26 has been rephrased as:

***$L_h(a)$ and $L_n(a)$ represents the space of hillslope and channel flow path from a grid-cell to the catchment outlet; K is the overland runoff velocity coefficient uses to distinguish hillslope routing to channel routing;***

10. L32, P6: Remove or replace "… using the parsimonious rainfall-runoff equation …"

We have removed the phrase (P7 line 8).

11. L18, P7: Rephrase "…in the mature and decade phase of the event …"

We have rephrased the sentence (P7 line 26-27) as:

*…but $[P]_a[W_1]_a$ overwhelms the other two in the mature and decaying phase of the event.*

12. L20, P7: Rephrase "…the differences start at negative …"

The phrase "*…the differences start at negative…*" has been changed to "*values of $[W_1]_a$-$[W_2]_a$ start negative…*" (P7 line 28).

13. L26, P7: Should read "… aggregated maps for terms in Eq. (8) …"

Done (P8 line 4).

14. L9, P8: Should read "…The magnitudes of terms in Eq.(9) …"

Done (P8 line 19).

15. L18, P8: Rephrase "…diminishing in magnitude with increase in scale is a result …" In the present form, this sentence is difficult to understand.

The sentence now reads as (P8 line 29-30):

"*…the diminishing in magnitude with increase in scale is a result of the decrease in catchment-average rainfall given that $W_2$ is constant among catchments.*"

16. L21, P8: Should read "…the subsurface component generally outperforms the surface one …"

Done (P8 line 32).

17. L3-4, P9: Not sure what the authors mean by "… a convex combination …", please explain or modify.

A convex combination of a probability distribution (often called a finite mixture distribution) refers to the weighted sum of its component probability distributions, with probability density function [see for example (Frühwirth-Schnatter, 2006)]:

$$f_R = \sum_i^N \psi_i f_{R_i}$$

where the following should hold:

$$\sum_{i}^{N} \psi_i = 1$$

In this study, it is trivia to show

$$\sum_{i}^{N} \psi_i = \sum_{i}^{N} \frac{[R_i]_{at}}{[R]_{at}} = \frac{\sum_{i}^{N}[R_i]_{at}}{[R]_{at}} = 1$$

Thus, $f_R$ (the PDF of catchment response time) is a convex combination of all $f_{Ri}$ (the PDF of catchment response time for component $i$). Perhaps we should keep the terms consistent by using either "distribution of catchment response time" or "PDF of catchment response time". We changed "***distribution***" to "***PDF***" in P9 line 11.

18. L10, P9: Should read "…where $|T_P|$ is the duration of the rainfall event."

Done (P9 line 17).

19. L2, P10: Should this be "… by rendering negative the mean E2 …"? Otherwise, this needs further clarification.

We have inserted "***values of***" before "***E2***" (P10 line 11).

20. L3-4, P11: Rephrase this sentence, in present form is confusing.

We modified the sentence (P11 line 9-10) as follows:

***As expected, runoff routing time (E(Θ)) increases according to catchment drainage area, which is attributed to the elongation in flow path.***

21. L5, P12: This is not needed and it is actually distracting.

We have deleted the sentence (P12 line 12).

22. L6, P12: Should read "… The major source of …"

Done (P12 line 16).

23. L20, P12: Not needed.

We have deleted the sentence (P13 line 5).

24. L16, P15: Should read "The middle panel of Figure 10 …"

Done (P16 line 13).

**References**

Frühwirth-Schnatter, S., 2006. Finite Mixture Modeling. In: Finite Mixture and Markov Switching Models. s.l.:Springer New York, pp. pp 1-23.

[revised manuscript text omitted]

---

## Author Response (AR1)

Hydrology and Earth System Science

Manuscript No.: hess-2016-255

Title: A Synthesis of Space-time Variability in Multi-Component Flood Response

Authors: Yiwen Mei, Xinyi Shen and Emmanouil N. Anagnostou

**NOTE:**

Reviewers' original comments are in 10pt Courier New type with yellow highlighting.

*Text from the revised manuscript is in bold italic font.*

Page/Line/Figure numbers given by reviewers were left unchanged in their text and refer to the originally submitted version of the manuscript

**Reply to comments from the Editor**

We would like to thank the Editor for his thorough and constructive comments. We have considered carefully all suggestions and comments and implemented revisions as noted below.

1. Firstly, I found it difficult to read and understand the presentation of the analytical framework without having to consult other papers. This was also pointed-out by the reviewers and something I think needs improving so this can be a standalone piece of work.

We were noticed of this issue and applied major changes through the entire manuscript. Specifically, please note our reply on major comment 4 of Reviewer 2. In addition, we added the derivations of quantities $E_i(T)$, $E_i(\Theta_i)$ and $E_i(T\Theta_i)$ in an Appendix instead of referring to V2010 paper. The changes are listed as follows:

*[P20, line 2 - 19] I. Catchment-average storm rainfall excess*

[revised manuscript text omitted]

==2. Section 4.1, line 3 it says that interception and Ea are not considered in this study, but they appear in Eq. (1) described as being part of the framework?==

Indeed they are not considered because the analytical framework is using through-rainfall as input. In Eq.(1) we describe how we derive the through-rainfall for this study. So, in applications of the analytical framework we would apply Eq.(1) on precipitation datasets using LAI and ET data. This is now clarified in the conclusions section.

*[P18, line 38 - P19, line 2] For instance, the vegetation interception can be estimated from the leaf area index data (Xiao, et al., 2014); database of the impervious area are provided in certain data rich locations (Homer, et al., 2015); and the spatial patterns of runoff coefficients could be retrieved in highly gauged catchments or from satellite-derived soil moisture fields at global scale (Merz & Blöschl, 2009; Penna, et al., 2011; Dhakal, et al., 2012; Massari, et al., 2014).*

==3. How is runoff determined from the impervious surfaces (W2)? Is it 100%?==

Yes, all through-rainfall (defined by Eq.(1)) that occurred over impervious surface is converted to rainfall excess. The impervious surface is spatially distributed parameter retrieved from the land use data. It could also be represented by a uniform imperviousness ($I_M$) parameter (averaged fraction of impervious surface for the catchment), which can be optimized through model calibration. In this study we chose the latter one for simplicity. To articulate this point, we modified the descriptions in text as shown below:

*[P4, line 32 - 33] In addition, the fraction of impervious surface in this study is represented by an imperviousness parameter that was optimized through model calibration.*

*[P6, line 10 - 14] The surface process is intimately related to the fraction of impervious surface over the basin where the through-rainfall is converted to rainfall excess, which can be represented as a uniform parameter, $I_M$, optimized through the hydrologic model calibration. Thus, the surface runoff coefficient, $W_2(a,t)$, is represented in the proposed framework by the imperviousness parameter, $I_M$.*

We also modified sentence for the subsurface runoff generation:

*[P6, line 15 - 16] …the amount of runoff generated from the subsurface is positively correlated to the soil wetness based on the variable infiltration curve adopted by CREST.*

==4. How should Eqs (6) and (7) be interpreted? The [*] and {*} notation used for expectation and covariance is, I think, confusing? How can an expectation be equal to a covariance?==

Eq.(6) means that the total rainfall excess is a sum of all the rainfall excess components generated by the different layers of subsurface. We revised the paragraph to make this clearly stated:

*[P7, line 11 - 14] Index i indicates different rainfall excess components generated from the different vertical layers of surface and subsurface. In this study we used two layers (i = 1 & 2) to denote the subsurface and surface rainfall excess, respectively. The total rainfall excess is the summation of all the rainfall excess components:*

Eq.(7) is the spatial expectation of Eq.(6). It shows that the basin-average rainfall excess time series are composed by the product of expectation between basin-average rainfall and runoff coefficient and the covariance between them. The notations for expectation and covariance are "*[ ]*" and "*{ }*", respectively. The sign "*\**" is used to represent the variable(s) that the operator is applied on. To get rid of the confusion, we removed "*\**" within "*[ ]*" and "*{ }*". Also, we separated the presentations for Eq.(7) and Eq.(8) to make the paragraph read more clearly. The paragraph reads as:

*[P7, line 15 - 19] To calculate the instantaneous basin-average rainfall excess, we take the spatial expectation of Eq.(6):*

$$[R]_a = [P]_a \sum_{i=1}^{N} [W_i]_a + \left\{ P, \sum_{i=1}^{N} W_i \right\}_a \qquad (7)$$

*where [ ]$_a$ and { }$_a$ stand for the expectation and covariance (variance if the variables are the same) operators applied over the catchment area. The distributed storm-average rainfall excess is given by taking the temporal expectation of Eq.(6):*

$$[R]_t = [P]_t \sum_{i=1}^{N} [W_i]_t + \left\{ P, \sum_{i=1}^{N} W_i \right\}_t \qquad (8)$$

5.   The paragraph from line 11 – 25 is very difficult to read and follow. Maybe try to add some more hydrological interpretation of what is going on?

We added further discussion on results and modified sentences in text as shown below:

*[P7, line 25 - 27] It is noted from Figure 4 that the catchment-average rainfall excess [R]$_a$ is strongly correlated to the catchment-average rainfall ([P]$_a$ shown in Figure 2) mainly because of the spatial covariance term {P,W$_1$}$_a$ that is irrelevant to [R]$_a$.*

*[P7, line 30 - P8, line 3] This is attributed to the dynamics of [W$_1$]$_a$ and [W$_2$]$_a$ during the event as shown by the differences between [W$_1$]$_a$ and [W$_2$]$_a$ on the top two panels of Figure 6. During the evolution of the event [W$_1$]$_a$-[W$_2$]$_a$ start negative and change to positive, reflecting the increase in subsurface runoff coefficient [W$_1$]$_a$ due to the increase in wetness condition of the catchment.*

*[P8, line 5 - 6] …this could be attributed to the lower maximum water capacity of the Swift catchment compared to the other two.*

We also included more details to explain the observations from Figure 5:

*[P8, line 10 - 13] This is exemplified for the Swift catchment due to its lower maximum water capacity. The temporal covariance between rainfall and subsurface runoff coefficient, {P,W₁}ₜ, is higher for the June 2006 event that exhibits more distinct rainfall bursts; and for the Swift catchment where W₁ is more sensitive to rainfall dynamics.*

6.   Page 8, line 28: fr is defined as the response time from start of a storm until a drop of water exists the catchment. Does this not then include both excess rainfall generation time and routing time? If yes, then how can the same pdf (fr) be used in the appendix to derive expected time for excess rainfall and routing time, respectively? Should each of these two processes not have their own specific pdf?

The catchment response time includes holding times from both rainfall excess generation and runoff routing. The random variables of rainfall excess generation time and runoff routing time share the same probability distribution (i.e. $f_R$) by definition of the analytical framework. Here is a rationale:

The catchment response time $\Phi$ is a random variable and it is the sum of $T$ and $\Theta$:

$$\Phi = T + \Theta$$

$f_R$ is the PDF of $\Phi$. So, the expectation of $\Phi$ is

$$E(\Phi) = \frac{1}{|T_P|}\int_{T_P}\frac{1}{|A|}\int_A (\varphi \cdot f_R)\,da\,dt$$

Substitute in $\Phi$ by $T$ plus $\Theta$, yield

$$E(\Phi) = E(T + \Theta)$$

$$= \frac{1}{|T_P|}\int_{T_P}\frac{1}{|A|}\int_A (t + \theta)\cdot f_R\,da\,dt$$

$$= \frac{1}{|T_P|}\int_{T_P}\frac{1}{|A|}\int_A (t \cdot f_R)\,da\,dt + \frac{1}{|T_P|}\int_{T_P}\frac{1}{|A|}\int_A (\theta \cdot f_R)\,da\,dt$$

$$= E(T) + E(\Theta)$$

Clearly, both $T$ and $\Theta$ take $f_R$ as the PDF.

We noted that the original manuscript did not specifically include a statement of the composition of $\Phi$. So, we added one sentence for this:

*[P9, line 8 - 9] The catchment response time is the sum of these two holding times and thus is also a random variable.*

7.   Eq. (16): When substituting Eq. (14) into (15) it seems like the psi disappears in (16), but maybe I am wrong?

Yes, we have also substituted $\psi_i$ by $[R_i]_{at}/[R]_{at}$ following Eq.(12). To make this clear, we modified sentence for Eq.(16) as:

*[P9, line 28] Substituting Eqs.(12) and (14) into Eq.(15)…*

Eq.(23) and the corresponding descriptions were also modified as:

*[P11, line 7] Substituting Eqs.(12) and (17) into Eq.(22), E(Θ) can be written as:*

$$E(\boldsymbol{\Theta}) = \underbrace{\frac{\sum_i^N \xi_i[\boldsymbol{\Theta}_i]_a[\boldsymbol{R}_i]_{at}}{[\boldsymbol{R}]_{at}}}_{E3} + \underbrace{\frac{\sum_i^N \xi_i\{\boldsymbol{\Theta}_i, [\boldsymbol{R}_i]_t\}_a}{[\boldsymbol{R}]_{at}}}_{E4} \qquad (23)$$

To make the forms of equations consistent, we also modified for the variance part of Eq.(27) and Eq.(30) as:

$$var(\boldsymbol{T}) = \underbrace{\frac{|\boldsymbol{T}_P|^2}{12}}_{v1} + \underbrace{\frac{\{\boldsymbol{T}^2, \sum_i^N[\boldsymbol{R}_i]_a\}_t - |\boldsymbol{T}_P|\{\boldsymbol{T}, \sum_i^N[\boldsymbol{R}_i]_a\}_t - \frac{\sum_i^N(\{\boldsymbol{T}, [\boldsymbol{R}_i]_a\}_t)^2}{[\boldsymbol{R}_i]_{at}}}{[\boldsymbol{R}]_{at}}}_{v2}$$

$$+ \underbrace{\sum_i^N \psi_i\left(\frac{\{\boldsymbol{T}, [\boldsymbol{R}_i]_a\}_t}{[\boldsymbol{R}_i]_{at}} - \frac{\{\boldsymbol{T}, \sum_i^N[\boldsymbol{R}_i]_a\}_t}{[\boldsymbol{R}]_{at}}\right)^2}_{LT} \qquad (27)$$

$$var(\boldsymbol{\Theta}) = \underbrace{\frac{\sum_i^N \xi_i^2\{\boldsymbol{\Theta}_i\}_a[\boldsymbol{R}_i]_{at}}{[\boldsymbol{R}]_{at}}}_{v3}$$

$$+ \underbrace{\frac{\sum_i^N \xi_i^2\{\boldsymbol{\Theta}_i^2, [\boldsymbol{R}_i]_t\}_a - 2\sum_i^N \xi_i^2[\boldsymbol{\Theta}_i]_a\{\boldsymbol{\Theta}_i, [\boldsymbol{R}_i]_t\}_a - \frac{\sum_i^N \xi_i^2(\{\boldsymbol{\Theta}_i, [\boldsymbol{R}_i]_t\}_a)^2}{[\boldsymbol{R}_i]_{at}}}{[\boldsymbol{R}]_{at}}}_{v4} \qquad (30)$$

$$+ \underbrace{\sum_i^N \psi_i\left(\xi_i[\boldsymbol{\Theta}_i]_a - \frac{\sum_i^N \xi_i[\boldsymbol{\Theta}_i]_a[\boldsymbol{R}_i]_{at}}{[\boldsymbol{R}]_{at}} + \frac{\xi_i\{\boldsymbol{\Theta}_i, [\boldsymbol{R}_i]_t\}_a}{[\boldsymbol{R}_i]_{at}} - \frac{\sum_i^N \xi_i\{\boldsymbol{\Theta}_i, [\boldsymbol{R}_i]_t\}_a}{[\boldsymbol{R}]_{at}}\right)^2}_{L\Theta}$$

**Reply to comments from Reviewer 1**

The paper provides a generalization of the analytical framework provided by Viglione et al. (2010), which in turns refers to Woods and Sivapalan (1999). With respect to Viglione et al. (2010) the authors introduce a multicomponent analysis of the flood streamflow but, on the other hand, in treating travel time-related issues they do not distinguish between hillslope and channel flow routing.

While the theoretical arguments and developments are interesting and well posed, I believe the paper should be reinforced with more attention to congruency with the hydrographs observed in real events.

In particular, in my opinion, more emphasis and major discussion should be provided about the comparison of the analytical results with the observed values of time to peak and hydrograph spreadness of the two observed events.

The weight and consequences of their assumption discarding the difference between hillslope and channel flow velocity should more deeply analyzed and discussed.

Results presented in Tables, if I correctly understood, are mean values referred to different observed events. Why not to show values referred to both the pilot events that look quite different for their distribution in time?

Obviously, the analytical framework requires data from a model for its application, nevertheless results and comments would be more acceptable if they were used also for a diagnostic analysis of the model itself.

The authors would like to thank the Anonymous Reviewer for rising these critical points in the original comment document. They are all very important aspects that can help to improve the scientific soundness and quality of the presentation of the manuscript. We have taken all comments and suggestions of the reviewer on board and prepared a detailed point-by-point response with applying the required modifications across the manuscript. We hope our response clears up the Reviewer's concern and strengthens our work.

Specifically, there are **four points** in the Reviewer's comments. For **point #1**, the Reviewer asked for more emphases and major discussion on comparison of the analytical framework results with the observation. We therefore included additional scatter plots of hydrograph properties derived from the analytical framework with respect to the observation in the original Figure 10. Corresponding statistics have also been added in the original Table 4. These resulted in major changes in discussions of the original Section 5 and paragraph 5 of Section 6.

[revised manuscript text omitted]

We did not add this information in the tables because this will make them very long (three times their current size). Also, since the values of these terms for the pilot events have already been shown in the figures, we believe adding this information in text alone should be sufficient.

The main aspect of **point #4** is the hydrologic representativeness of the CREST simulation results, which in turn refers to also the representativeness of the analytical framework. To articulate this point, we refer to our discussion in lines 34 - 37 of page 4 and lines 6 - 8 of page 5 for the performance of the simulations. We also added content to describe the values of error metrics of the two pilot events in text as:

*[P5, line 9 - 11] They are characterized by high CC and low relative CRMS values with respect to the observed flow time series. The CC for the two events are above 0.94 and 0.82 with relative CRMS at about 50% (Swift catchment for event 2 is an exception with CRMS at about 100%).*

**Reply to comments from Reviewer 2**

**Major comments**

1. The introduction of the paper is complete and frames the problem interestingly. I have not additional comments there.

Thank you.

2. The manuscript requires additional editorial work. Below I indicate some edits that I was able to identify, but the authors need to make sure that typos and grammatical mistakes are completely resolved.

Thank you. We have further edited the manuscript.

3. I'm confused about the number and the basis for choosing the multi-components of the flood response. In the introduction section, it seems the components are chosen based on different land use classes (see, e.g., lines 35-37, P2). Then, in lines 18-20, P5, it appears that only two components are considered and on the basis of flow paths, namely surface or subsurface, rather than land cover classes. This requires clarification.

This is a very good point and it definitely needs to be clarified. The separation of rainfall excess into components is based on the heterogeneity of vertical soil layers. Each of the sub-surface layers is associated with a flow path. This is typically represented in distributed hydrologic models implicitly using the multiple reservoirs approach, or in more advanced models through explicit equations. Land cover types are then used to characterize the distribution of the excess rainfall over the vertical soil layers. For example, for an urbanized land cover a large portion of the excess rainfall is located within the very shallow layer (the first layer) of the soil while there is a shift of the excess rain water from the shallow to the deeper layers for more vegetated land cover types. To articulate these aspects, we have applied several modifications throughout the manuscript.

*[P1, line 2 - 4] Catchment flood response consists of multiple components of flow originating from different surface and sub-surface layers. This study proposes an extension of Viglione et al. (2010a) analytical framework to represent the dependence of catchment flood response to the different runoff generation processes.*

*[P2, line 28 - 32] In this sense, the different runoff generation processes associated with vertical heterogeneous catchment layers are lumped together into a single flood response (Woods & Sivapalan, 1999; Viglione, et al., 2010b). Numerous experimental studies, though, have demonstrated that catchment flood response can be identified as multiple components originating from different catchment layers and associated with different flow paths (Weiler, et al., 2003; Liu, et al., 2004; Gonzales, et al., 2009).*

*[P2 line 36 - 38] Catchment rainfall forcing is converted to more than one rainfall excess components associated with various surface and subsurface layers. These rainfall excess components are subjected to different flow paths and routing schemes.*

*[P18, line 27 - 30] This is particularly useful in analyzing the flood response of urbanized catchments where the distribution of excess rainfall into different vertical soil layers is quite different between the highly impervious urban areas (e.g. roads, rooftops, parking lots, etc.) and the more pervious suburban or rural areas of the basin (Smith, et al., 2002; Mejía & Moglen, 2010; Mejía, et al., 2015).*

4.  In some instances, the manuscript refers the reader to equations in another published paper, i.e. V2010. This should be avoided to make the manuscript readable and self-contained. For example, this occurs in lines 3-5, P8, but in other places as well.

Thank you, we see the point here. We provided the meaning of each term instead of referring the readers to V2010. There are six modifications reflected in the manuscript.

*[P7, line 11] The generation of rainfall excess at location and time (a,t) is calculated as:*

*[P8, line 18 - 21] Term R1 represents the sum of product between the catchment-average storm rainfall and runoff coefficient for all components. Term R2/R3 is the sum of temporal/spatial covariance between the catchment-/storm-average rainfall and runoff coefficient. R4 is the sum of temporal covariance between spatial variation of precipitation and runoff coefficient.*

*[P10, line 1 - 2] E1 refers to the event half-duration and E2 is the expectation of time distance from the event midpoint to the temporal mass center of catchment-average rainfall excess.*

*[P11, line 8 - 9] E3 stands for the spatial mean of runoff routing time in the hillslope and channel network. E4 is the expected distance from the geomorphologic center of catchment to the centroid of storm-average rainfall excess.*

*[P12, line 17 - 20] Term v1 stands for the variance in time generated by a temporal invariant catchment-average rainfall excess. Term v2 represents component-wised mean of additional variance caused by the temporal variation in catchment-average rainfall excess. The last term is named LT and represents the mean square of "time lag" (between each component to the total) in rainfall excess generation.*

*[P13, line 21 - 23] Term v3 represents the variance in time generated by a spatial invariant storm-average rainfall excess. Term v4 is the mean of additional variance caused by the spatial variation in storm-average rainfall excess. The term LƟ represents the mean of "time lag" in runoff routing between rainfall excess components to the total.*

In addition to these changes, we also referred to our reply on comment 1 from the editor on this issue.

5.  The conclusion section could be organized using bullet points. Additionally, it should make clear (e.g., line 28. P16) that they are based on the selected

study basins.

Thanks for the suggestion but we choose to go with paragraph instead of bullet points.

**Minor comments**

1.  L36, P2: Should read "… excess component by the various …"

Done (P2, line 37).

2.  L42, P2: Should read "… the most dominant for a component …"

Done (P3 line 1).

3.  L16-17, P3: "The readers are encouraged to read Mei & Anagnostou (2015) for details on the hydrology of the study area", consider rephrasing to, e.g., "The reader is refer to Mei & Anagnostou (2015) for further details on the hydrology of the study area."

The sentence has been modified as:

*[P3, line 17 - 18] The reader is referred to Mei & Anagnostou (2015) and Mei et al. (2014) for details on the hydrology of the study area.*

4.  L24-25, P3: Rephrase "Another meteorology data is the potential evapotranspiration (PET) data available …"

The sentence has been rephrased as:

*[P3, line 25 - 27] Another atmospheric forcing dataset used in this study is the potential evapotranspiration (PET) available from the North American Regional Reanalysis (NARR) at 3-hourly and 32 km resolution (Mesinger et al., 2006).*

5.  L28-29, P3: Should read "… record available from the United States Geological Survey (USGS) …"

Done (P3, line 30).

6.  L32, P3: Should read "…are its parsimonious data requirement …"

Done (P3 line 33).

7.  L11, P4: It would help to include here the spatial resolution of the CREST model used in this study.

We reported the model grid domain resolution in text as:

*[P4, line 26] …the model is set up over the three study catchments with 1 km spatial resolution.*

8.  L23, P4: Should read "… parameter optimization algorithm adopted in CREST …"

We removed the sign "–".

The sentence has been rephrased as:

*[P6, line 27 - 29] $L_h(a)$ and $L_n(a)$ represents the space of hillslope and channel flow path from a grid-cell to the catchment outlet; K is the overland runoff velocity coefficient uses to distinguish hillslope routing to channel routing;*

10.  L32, P6: Remove or replace "… using the parsimonious rainfall-runoff equation …"

We removed the phrase.

11.  L18, P7: Rephrase "…in the mature and decade phase of the event …"

We rephrased the sentence as:

*[P7, line 30] …but $[P]_a[W_1]_a$ overwhelms the other two in the mature and decaying phase of the event.*

The sentence now read as:

*[P8, line 1 - 2] During the evolution of the event $[W_1]_a$-$[W_2]_a$ start negative and change to positive…*

The sentence in the original manuscript is the same as the one suggested by the reviewer.

The sentence in the original manuscript is the same as the one suggested by the reviewer.

15.  L18, P8: Rephrase "…diminishing in magnitude with increase in scale is a result …" In the present form, this sentence is difficult to understand.

The sentence now reads as:

*[P8, line 33 - 34] …the diminishing in magnitude with increase in scale is a result of the decrease in catchment-average rainfall given that $W_2$ is constant among catchments.*

16.  L21, P8: Should read "…the subsurface component generally outperforms the surface one …"

Done (P9 line 1).

17.  L3-4, P9: Not sure what the authors mean by "… a convex combination …", please explain or modify.

A convex combination of a probability distribution (often called a finite mixture distribution) refers to the weighted sum of its component probability distributions, with probability density function [see for example (Frühwirth-Schnatter, 2006)]:

$$f_R = \sum_i^N \psi_i f_{R_i}$$

where the following should hold:

$$\sum_i^N \psi_i = 1$$

In this study, it is trivia to show

$$\sum_i^N \psi_i = \sum_i^N \frac{[R_i]_{at}}{[R]_{at}} = \frac{\sum_i^N [R_i]_{at}}{[R]_{at}} = 1$$

Thus, $f_R$ (the PDF of catchment response time) is a convex combination of all $f_{Ri}$ (the PDF of catchment response time for component $i$). Perhaps we should keep the terms consistent by using either "distribution of catchment response time" or "PDF of catchment response time". We changed "*distribution*" to "*PDF*" in P9 line 15.

18.  L10, P9: Should read "…where |TP| is the duration of the rainfall event."

Done (P9 line 21).

19.  L2, P10: Should this be "… by rendering negative the mean E2 …"? Otherwise, this needs further clarification.

We inserted "*values of*" before "*E2*" (P10 line 13).

20.  L3-4, P11: Rephrase this sentence, in present form is confusing.

We modified the sentence as follows:

*[P11, line 12 - 13] As expected, runoff routing time (E(Θ)) increases according to catchment drainage area, which is attributed to the elongation in flow path.*

21.  L5, P12: This is not needed and it is actually distracting.

We deleted the sentence.

22.  L6, P12: Should read "… The major source of …"

Done (P12, line 18).

23.  L20, P12: Not needed.

We deleted the sentence.

24.  L16, P15: Should read "The middle panel of Figure 10 …"

Done (P16, line 16).

**References**

Frühwirth-Schnatter, S., 2006. Finite Mixture Modeling. In: Finite Mixture and Markov Switching Models. s.l.:Springer New York, pp. pp 1-23.

[revised manuscript text omitted]

$$\psi_i = \frac{[R_i]_{at}}{[R]_{at}} \tag{12}$$

Sum of $\psi_i$ goes up to 1 by definition. Eq.(11) shows that the PDF of catchment response time is a convex combination for each PDF of the rainfall excess component.

**4.3.1. Expectation of Catchment Response Time**

For the two-stage analytical framework in this study, the expectation of catchment response time $E(\Phi)$ can be decomposed to the expectation of holding time of the two stages:

$$E(\Phi) = \underbrace{E(T)}_{Stage1} + \underbrace{E(\Theta)}_{Stage2} \tag{13}$$

where $T$ and $\Theta$ correspond to the rainfall excess generation time and runoff routing time. The expected rainfall excess generation time, $E_i(T)$, for any component is provided as (see Appendix II):

$$E_i(T) = \frac{|T_P|}{2} + \frac{\{T, [R_i]_a\}_t}{[R_i]_{at}} \tag{14}$$

where $|T_P|$ is the duration of the rainfall event. $E_i(T)$ is a measurement of the temporal mass center of rainfall excess. If the rainfall mass is symmetric with respect to its mid-point, the half-duration is sufficient to describe the expectation of rainfall excess generation. Following the distribution function of Eq.(11), we derived the expected rainfall generation time for total rainfall excess $E(T)$ as (see Appendix II for the derivation):

$$E(T) = \sum_i^N \psi_i E_i(T) \tag{15}$$

Eq.(15) indicates that the temporal mass center of total rainfall excess is a linear combination (or the expectation) of the mass centers of all the other rainfall excess components with respect to the rainfall excess ratio. The equation also implies that the larger the magnitude of a component, the greater impact it has on the timing of the total rainfall excess. Substituting Eqs.(12) and (14) into Eq.(15), we have,

Commented [MY31]: Reviewer 2, minor comment 16

Commented [MY32]: Editor comment 6.

Commented [MY33]: Reviewer 2, minor comment 17

Commented [MY34]: Reviewer 2, minor comment 18

Commented [MY35]: Editor comment 7

[revised manuscript text omitted]

**Commented [MY69]:** Reviewer 1, point 1